# Hsp70 in Liquid Biopsies—A Tumor-Specific Biomarker for Detection and Response Monitoring in Cancer

**DOI:** 10.3390/cancers13153706

**Published:** 2021-07-23

**Authors:** Caroline Werner, Stefan Stangl, Lukas Salvermoser, Melissa Schwab, Maxim Shevtsov, Alexia Xanthopoulos, Fei Wang, Ali Bashiri Dezfouli, Dennis Thölke, Christian Ostheimer, Daniel Medenwald, Martin Windberg, Matthias Bache, Martin Schlapschy, Arne Skerra, Gabriele Multhoff

**Affiliations:** 1Center for Translational Cancer Research (TranslaTUM), Radiation Immuno-Oncology Group, Technical University of Munich (TUM), Klinikum rechts der Isar, Einsteinstr. 25, 81675 Munich, Germany; c.werner@tum.de (C.W.); stefan.stangl@tum.de (S.S.); lukas.salvermoser@gmx.de (L.S.); melissa.schwab@tum.de (M.S.); maxim.shevtsov@tum.de (M.S.); alexia.x@gmx.de (A.X.); fei.wang@tum.de (F.W.); ali.bashiri@tum.de (A.B.D.); dennis.thoelke@hotmail.de (D.T.); 2Department of Radiation Oncology, Technical University of Munich (TUM), Klinikum rechts der Isar, Ismaningerstr. 22, 81675 Munich, Germany; 3Institute of Cytology of the Russian Academy of Sciences (RAS), Tikhoretsky Ave. 4, 194064 Saint Petersburg, Russia; 4Department of Radiation Therapy, University Hospital Halle/Saale, Ernst-Grube Str. 40, 06120 Halle, Germany; christian.ostheimer@uk-halle.de (C.O.); daniel.medenwald@uk-halle.de (D.M.); martin.windberg@student.uni-halle.de (M.W.); matthias.bache@uk-halle.de (M.B.); 5Lehrstuhl für Biologische Chemie, School of Life Sciences, Technical University of Munich (TUM), 85354 Freising, Germany; schlapschy@wzw.tum.de (M.S.); skerra@tum.de (A.S.)

**Keywords:** Hsp70, sandwich ELISA, liquid biopsy, tumor biomarker, small extracellular vesicles, prediction, response monitoring, non-small cell lung carcinoma (NSCLC), glioblastoma

## Abstract

**Simple Summary:**

Tumor-specific biomarkers in liquid biopsies provide useful tools for detection of tumors, monitoring of tumor responses and prediction of outcomes. Nearly all malignant solid tumor cells, but not normal cells, present the major stress-inducible Heat shock protein 70 (Hsp70) on their cell surface and actively release it into the blood in small extracellular vesicles. Therefore, vesicular Hsp70 might serve as a biomarker for viable tumor cells. Presently, no validated test system is available that allows the quantification of vesicular Hsp70 in the blood. Based on two Hsp70-specific monoclonal antibodies, we have developed the complete (comp)Hsp70 ELISA that provides a highly sensitive and reliable tool for measuring both, free and vesicular Hsp70 in the circulation of tumor patients. Hsp70 levels in the blood reflect the presence and risk characteristics of tumors and their membrane-Hsp70 status, and might be predictive for therapeutic responses.

**Abstract:**

In contrast to normal cells, tumor cells of multiple entities overexpress the Heat shock protein 70 (Hsp70) not only in the cytosol, but also present it on their plasma membrane in a tumor-specific manner. Furthermore, membrane Hsp70-positive tumor cells actively release Hsp70 in small extracellular vesicles with biophysical characteristics of exosomes. Due to conformational changes of Hsp70 in a lipid environment, most commercially available antibodies fail to detect membrane-bound and vesicular Hsp70. To fill this gap and to assess the role of vesicular Hsp70 in circulation as a potential tumor biomarker, we established the novel complete (comp)Hsp70 sandwich ELISA, using two monoclonal antibodies (mAbs), that is able to recognize both free and lipid-associated Hsp70 on the cell surface of viable tumor cells and on small extracellular vesicles. The epitopes of the mAbs cmHsp70.1 (aa 451–461) and cmHsp70.2 (aa 614–623) that are conserved among different species reside in the substrate-binding domain of Hsp70 with measured affinities of 0.42 nM and 0.44 nM, respectively. Validation of the compHsp70 ELISA revealed a high intra- and inter-assay precision, linearity in a concentration range of 1.56 to 25 ng/mL, high recovery rates of spiked liposomal Hsp70 (>84%), comparable values between human serum and plasma samples and no interference by food intake or age of the donors. Hsp70 concentrations in the circulation of patients with glioblastoma, squamous cell or adeno non-small cell lung carcinoma (NSCLC) at diagnosis were significantly higher than those of healthy donors. Hsp70 concentrations dropped concomitantly with a decrease in viable tumor mass upon irradiation of patients with approximately 20 Gy (range 18–22.5 Gy) and after completion of radiotherapy (60–70 Gy). In summary, the compHsp70 ELISA presented herein provides a sensitive and reliable tool for measuring free and vesicular Hsp70 in liquid biopsies of tumor patients, levels of which can be used as a tumor-specific biomarker, for risk assessment (i.e., differentiation of grade III vs. IV adeno NSCLC) and monitoring of therapeutic outcomes.

## 1. Introduction

Lung cancer is a major cause of cancer-related deaths and the second most common cancer in men and women, worldwide [1]. Due to its nonspecific symptoms, lung cancer is frequently diagnosed at a late disease stage [2]. A relevant proportion of patients with locally advanced or metastasized tumors does not show an improvement in progression-free and overall survival following radical surgery, simultaneous chemo- and radiotherapy or/and immune checkpoint inhibitors [3,4]. Like NSCLC, glioblastoma multiforme (GBM) is a devastating disease of the central nervous system with symptoms that present at a late disease stage. Despite multimodal treatment strategies consisting of surgery, radiotherapy and a temozolomide-based chemotherapy, overall survival remains poor at 15–18 months [5]. These examples underline the high medical need for tumor-specific biomarkers that improve the detection of tumors and the monitoring of therapeutic responses. The development of such biomarkers in liquid biopsies will increase therapeutic success and the life expectancy of patients with highly aggressive tumors, since blood samples can be taken repeatedly and thereby monitoring of tumor responses—as determined by a drop/increase in circulating Hsp70—can be intensified which would result in a faster therapy adaptation. Another challenge in clinical practice is the potential side effects of image-guided medical diagnosis. The availability of a minimally invasive method such as blood sampling for assessing the presence of tumor-specific biomarkers in liquid biopsies will have a broad applicability and will be well tolerated. Herein, we present an ELISA-based quantification of free and vesicular Hsp70 (HSPA1A) as a reliable approach for detecting tumors and monitoring therapeutic responses.

Members of the 70 kDa chaperone family support folding of nascent polypeptides, prevent protein aggregation and assist transport of proteins across membranes [6,7], and they reside in nearly all subcellular compartments of nucleated cells [8]. The importance of Hsp70 is documented by its high abundance, evolutionary conserved amino acid (aa) sequence [9,10] and functional similarities such as maintenance of protein homeostasis across different species [11]. Transgenic rodent models have revealed that Hsp70 of *Drosophila melanogaster* can substitute the activity of murine Hsp70 [12,13,14], and human Hsp70 expressed in myocardial cells of transgenic rats can protect the heart from ischemic stress in vivo [15].

In contrast to normal cells, tumor cells frequently overexpress the major stress-inducible Hsp70 [16] in the cytosol and present it on the plasma membrane in a tumor-specific manner [17]. A global profiling of cell surface-bound proteins revealed a high abundancy of HSP70 and other intracellular chaperone families, such as GRP78, HSP60, HSP54 and HSP27, on the plasma membrane of different tumor cells [18]. It is assumed that Hsp70 trafficking to the plasma membrane is enabled by a non-ER/Golgi, alternative endo-lysosomal pathway [19]. Since major changes in extracellular salt concentrations and pH fail to deplete Hsp70 from the plasma membrane, a (trans-) membrane receptor mediated anchorage of Hsp70 is highly unlikely. Lipid profiling and artificial lipid copellation assays revealed that Hsp70 directly interacts with glycosphingolipids such as globoyltriaosylceramide (Gb3), which localize in cholesterol-rich microdomains (rafts) in the membrane of tumor cells [19]. Since normal cells lack these tumor-specific lipids in their plasma membrane, Hsp70 resides strictly in the cytosol of normal cells, and small extracellular vesicles released from normal cells remain membrane Hsp70-negative. Following stress such as sublethal heat or cytostatic drugs the synthesis of Hsp70 increases also in normal cells, but Hsp70 does not translocate to the plasma membrane, which is a prerequisite for Hsp70-positive extracellular vesicles [20]. In tumor cells, stress triggers an interaction of Hsp70 with the apoptosis-related membrane lipid component phosphatidylserine (PS). Based on its negative charge PS enables an insertion of Hsp70 in artificial lipid vesicles [21,22].

An Hsp70 membrane positivity has been found in numerous different tumor entities [23], including lung, head and neck, colorectal, pancreas, breast carcinomas and hematological malignancies [24,25]. Tumor cells presenting Hsp70 on their plasma membrane are more resistant to radiotherapy and chemotherapy compared to their membrane Hsp70-negative counterparts [26]. After exposure to environmental stress, the synthesis and membrane expression of Hsp70 is further upregulated in tumor cells. A high Hsp70 content contributes to an aggressive tumor phenotype, mediates protection against apoptosis, promotes invasion/migration and mediates resistance to standard therapies [27]. Mouse models revealed that metastases exhibit a higher membrane Hsp70 density than primary tumors [28] and pre-malignant lesions, and early stage tumors show a lower membrane Hsp70 expression than malignant esophageal adeno carcinoma (Ms in resubmission). Moreover, viable tumor cells expressing Hsp70 on their plasma membrane actively release small extracellular vesicles with biophysical properties of exosomes, whereas free Hsp70 predominantly originates from dying cells [29,30,31]. Since small extracellular vesicles are created by a double invagination, the protein content on their surfaces reflects that of the tumor cell membrane from which they originate [32]. As a result, membrane Hsp70-positive tumor cells release extracellular vesicles presenting Hsp70 on their surface [31], and the lumen of these vesicles contains proteins of the tumor cytosol, but not of the ER [33].

We have previously shown that circulating Hsp70 levels in patients with tumors are significantly higher than in patients with inflammation or healthy donors [34]. Therefore, based on the Hsp70 levels in the circulation, inflammatory diseases can be distinguished from cancer. A high Hsp70 serum content correlates with an increased malignancy and resistance to chemo- and radiotherapy [25,35,36,37]. The precise detection of the viable tumor mass requires the measurement of vesicular Hsp70, which is actively released by viable tumor cells. Commercially available Hsp70 ELISA systems only detect free Hsp70 in aqueous solutions, but not vesicular Hsp70 in serum or plasma because an interaction of Hsp70 with lipids i.e., PS, Gb3, sphingolipids induces conformational changes in Hsp70 [38,39]. The novel compHsp70 sandwich ELISA is based on two mAbs, cmHsp70.1 and cmHsp70.2, recognizing conserved epitopes in the C-terminal substrate binding domain of inducible Hsp70 [40,41] and allowing a specific and sensitive quantification of both, free and vesicular Hsp70 in the blood of patients with cancer.

## 2. Materials and Methods

If not indicated otherwise all reagents were obtained from Sigma-Aldrich, St Louis, MO, USA.

### 2.1. Uniprot Analysis

Inter-species comparison of the 8-mer (aa 454–461) and 10-mer (aa 614–623) sequences of Hsc70 (HSPA8) and Hsp70 (HSPA1A) in humans were compared to the respective sequence of Hsp70 in mouse, rat, dog, bovine, horse, pig and zebrafish by UniProtKB 2021_03 analysis. For cats, only the sequence of HSPA2 is available.

### 2.2. Recombinant Hsp70

Human recombinant Hsp70 protein was produced in an optimized SF9 insect cell line (Orbigen, San Diego, CA, USA). Briefly, SF9 cells were transfected with baculovirus carrying the cDNA encoding for human Hsp70, which was fused with an N-terminal poly-His-tag (Orbigen). After transfection, His-tagged Hsp70 was isolated from cell lysates by loading on a His-Trap nickel sepharose column using an Äkta Purifier liquid chromate-graphy system (GE Healthcare, Chalfont St. Giles, UK). His-tagged Hsp70 was eluted by increasing imidazole concentrations in a saline buffer (20 mM sodium phosphate, 0.5 M NaCl, 0.5 M imidazole, pH 7.4; all Sigma-Aldrich) and pooled from different fractions. After subsequent desalting and re-buffering using PD-10 columns (GE Healthcare) the Hsp70 protein yield was determined using a bichinchoninic acid (BCA) protein kit (Pierce, Thermo Fisher Scientific, Rockford, IL, USA). Hsp70 aliquots of 50 µg/mL were stored at −80 °C.

### 2.3. Microscale Thermophoresis-Based Affinity Measurements

Binding affinity of the rodent cmHsp70.1 and cmHsp70.2 monoclonal antibodies (mAbs) to Hsp70 protein was determined using the Microscale Thermophoresis (MST) technique [42,43]. For MST measurements, gradual thermophoretic changes of a constant concentration of FITC-labeled cmHsp70.1 and cmHsp70.2 mAbs (multimmune GmbH, Munich, Germany) were incubated for 10 min with rising concentrations of recombinant Hsp70 protein (range 0.000148 nM to 11.8 nM) and analyzed on the Monolith NT.115 (NanoTemper Technologies GmbH, Munich, Germany).

### 2.4. Western Blot and Dot Blot Analysis

Tumor cells were lysed in radioimmunoprecipitation assay (RIPA)-buffer containing 50 mM Tris-HCl (pH 8.0), 150 mM NaCl, 1 mM EDTA, 1% *v*/*v* Triton-x-100, 0.1% *w*/*v* sodium dodecyl sulfate (SDS), 0.5% *w*/*v* sodium deoxycholate (all Sigma-Aldrich) and protease inhibitor cocktail (Roche, Basel, Switzerland). The protein content was determined using the BCA protein assay kit (Thermo Fisher Scientific, Waltham, MA, USA). Equal amounts of recombinant Hsp70 protein (50 ng) as well as lysates of U87 glioblastoma cells (50 µg) were subjected to SDS-PAGE, transferred to nitrocellulose membranes (GE Healthcare Life Sciences, Chicago, IL, USA), blocked in 5% *w*/*v* skimmed milk and incubated with the following Hsp70 antibodies (4.8 µg/mL, each): cmHsp70.1 (multimmune GmbH), cmHsp70.2 (multimmune GmbH), hHSP70/HSPA1A Clone 998953 (R&D Systems, Inc., Minneapolis, MN, USA) and hHSP70/HSPA1A Clone 242707 (R&D Systems, Inc.) and β-actin (A2228; Sigma-Aldrich). As secondary antibodies, horseradish-peroxidase (HRP)-conjugated anti-mouse (P0260, 1:2000; Dako-Agilent, Santa Clara, CA, USA) and anti-rat (P0450; 1:1000; Dako-Agilent) antibodies were used. For dot blot analysis, different amounts (100 ng, 10 ng, 1 ng 0.1 ng in a volume of 2 µL PBS; Life Technologies, Carlsbad, CA, USA) of BSA (Sigma-Aldrich) and HSP proteins (Hsp27, SPR-118; Hsp60, SPR-104; Hsp70, SPR-103; Hsc70, SPR-106; StressMarq Biosciences Inc., Victoria BC, Canada) were spotted on nitrocellulose membranes and incubated with the antibodies, as described above. Immune complexes were detected by Pierce ECL Western Kit (Thermo Fisher Scientific) and imaged digitally (ChemiDoc Touch Imaging System, BioRad, Hercules, CA, USA). Fiji software was used for quantification of the Western blot signals.

### 2.5. Peptide SPOT Synthesis and Analysis

An array of consecutive 14-mer peptides with 12-residue overlap, covering the amino acid (aa) sequence 382–641 of human Hsp70, was synthesized according to the SPOT method, as previously described [44,45], on a Gly-PEG500-derivatised cellulose membrane using a MultiPep RS instrument (Intavis, Cologne, Germany). After N-terminal acetylation and de-protection of the peptide side chains, the membrane was treated with Membrane-Blocking Solution (MBS) containing 10% *v*/*v* Western blocking reagent (Roche Diagnostics, Mannheim, Germany) and washed with MBS. After incubation with 5 µg/mL cmHsp70.2 in MBS for 1 h at room temperature and washing with MBS, an incubation with a HRP-conjugated anti-rat antibody (P0450; 1:1000 in MBS; Dako-Agilent) was performed, followed by a staining with 3,3′-diaminobenzidine (SigmaFAST^TM^, Sigma-Aldrich).

### 2.6. Cell Culture

The human lung carcinoma cell line A549 (adenocarcinoma alveolar basal epithelial cells, ATCC^®^ CCL-185™) was cultured in RPMI1640 Medium (Sigma-Aldrich) and the human colon carcinoma cell line HCT116 (ATCC^®^ CCL-247™) was cultured in Dulbecco’s Modified Eagle’s Medium (DMEM; Sigma-Aldrich), both supplemented with 10% heat-inactivated fetal bovine serum (FBS; Sigma-Aldrich), 2 mM L-glutamine (Sigma-Aldrich), 1 mM sodium pyruvate and antibiotics (100 IU/mL penicillin and 100 mg/mL streptomycin; Sigma-Aldrich) and cultured at 37 °C with 5% *v*/*v* CO_2_ in a humidified atmosphere. Tumor cell lines were routinely monitored for mycoplasma contaminations (MycoAlert Mycoplasma Detection Kit; Lonza, Basel, Switzerland), and only mycoplasma-negative cells were used for analysis. All experiments were performed in the exponential growth phase on day 2 after seeding.

### 2.7. Flow Cytometry

Cells (300,000) were harvested, washed twice in ice cold flow cytometry buffer (phosphate buffered saline (PBS; Life Technologies, containing 10% *v*/*v* FBS, Sigma-Aldrich) and incubated with the FITC-labeled cmHsp70.1, cmHsp70.2, hHSP70/HSPA1A Clone 998953 and hHSP70/HSPA1A Clone 242707 for 30 min in the dark on ice. Prior to acquisition, unbound antibody was removed by a second washing step. Propidium iodine (PI, Merck, Darmstadt, Germany) was added (1 μg/mL) directly before analysis and only viable (PI-negative) cells gated upon and analyzed on a FACSCalibur™ flow cytometer (BD Biosciences, Franklin Lakes, NJ, USA). Isotype-matched control antibodies were used as respective negative controls.

### 2.8. Hsp70 Containing Artificial Lipid Vesicles

Unilamellar artificial lipid microvesicles were freshly prepared for each experiment and used as ‘model’ for small extracellular vesicles. Briefly, 1-palmitoyl-2-oleoyl-sn-glycero-3-phosphocholine (POPC) and 1-palmitoyl-2-oleoyl-sn-glycero-3-phospho-L-serine (POPS; both Avanti Polar Lipids, Alabaster, AL, USA), dissolved in chloroform, were mixed in a molar ratio of 8:2 and dried under a stream of nitrogen gas. Dried lipids were dissolved in 25 mM Tris/HCl pH 7.4 and 250 mM NaCl (1 mL/10 mg of lipid) and incubated for 1.5 h at room temperature. The lipid suspension was forced through a lipid extruder (Avanti Polar Lipids, Alabaster, AL, USA) equipped with a 100 nm polycarbonate filter 15 times in order to generate unilamellar artificial liposomes (final concentration 1 mg/mL in 1 mM Bis/Tris buffer, pH 7.4, Sigma-Aldrich). Recombinant Hsp70 (1 mg/mL) was incubated with the lipid solution for 30 min at room temperature. After adding of ultrapure H_2_O, Hsp70-containing lipids were ultra-centrifuged at 200,000× *g* at 4 °C for 2 h and pellets were subsequently resuspended in PBS (Life Technologies). Hsp70-containing and Hsp70-free control lipid vesicles were used within 24 h after preparation. The presence of Hsp70 protein in the pellet fraction, but not in the supernatant, confirmed the binding of Hsp70 to lipid vesicles. The size and uniformity of the lipid vesicles were analyzed by dynamic light scattering (Zetasizer NanoS; Malvern Instruments, Malvern, UK).

### 2.9. Collection of Human Plasma and Serum Samples

Blood samples (7.5 mL each) were taken from healthy donors (*n* = 108), and patients with non-small cell lung cancer (NSCLC; *n* = 166) and glioblastoma multiforme (*n* = 34). Blood samples were also collected from patients with lung cancer at diagnosis (*n* = 80), during radiotherapy (after 20 Gy; *n* = 58) and after finishing radiotherapy (after 60–70 Gy; *n* = 56). All study participants provided informed, written consent. Approval of the study was obtained by the local ethical committees of the Klinikum rechts der Isar, Technical University of Munich. Plasma was prepared from EDTA blood (S-Monovette, Sarstedt, Nümbrecht, Germany) by centrifugation at 1500× *g* for 15 min at room temperature. Serum was obtained after clotting of the blood for 30 min at room temperature in a serum separator tube with clotting activator (S-Monovette, Sarstedt, Nümbrecht, Germany), followed by centrifugation at 750× *g* for 10 min. Serum and plasma were stored in aliquots (150 μL) at −80 °C. To test the influence of food intake as a potential interference factor, serum samples were collected from healthy human individuals before and 2 h after intake of a high fat diet.

### 2.10. Biotinylation of the Detection Antibody

The cmHsp70.1 monoclonal antibody and its corresponding control antibody, hHSP70/HSPA1A (Clone 242707), were biotinylated using EZ-link sulfo-NHS-LC-biotin (Thermo, Rockford, IL, USA). After adjustment of the carbonate-buffer to pH 8.5, the antibody solution was incubated with a 40-fold molar excess of NHS-LC-biotin for 1 h at room temperature. The remaining free biotin was removed using Zeba spin desalting columns (Thermo Fisher Scientific, Waltham, MA, USA). Protein concentrations of the antibodies were determined using the BCA protein kit (Pierce, Thermo, Rockford, IL, USA) following the manufacturer’s recommendations.

### 2.11. compHsp70 Sandwich ELISA

96-well MaxiSorp Nunc-Immuno plates (Thermo, Rochester, NY, USA) were coated overnight by incubating with 1 μg/mL cmHsp70.2 (multimmune GmbH) in sodium carbonate buffer (0.1 M sodium carbonate, 0.1 M sodium hydrogen carbonate, pH 9.6; Sigma-Aldrich). After washing with PBS (Life Technologies) supplemented with 0.05% Tween-20 (Calbiochem, Merck, Darmstadt, Germany), nonspecific binding was blocked by incubation with liquid plate sealer (Candor Bioscience GmbH, Wangen i. Allgäu, Germany) for 30 min at room temperature. Following another washing step, serum or plasma samples diluted in StabilZyme Select Stabilizer (Diarect GmbH, Freiburg i. Breisgau, Germany), were added to the wells and incubated for 30 min at room temperature. An eight-point concentration standard curve of Hsp70 protein (0–100 ng/mL) diluted in StabilZyme Select Stabilizer (Diarect GmbH, Freiburg i. Breisgau, Germany) was included in each assay. After another washing step, wells were incubated with 200 ng/mL of the biotinylated cmHsp70.1 (multimmune GmbH, Munich, Germany) in HRP-Protector (Candor Bioscience GmbH, Wangen i. Allgäu, Germany) for 30 min at room temperature. Following a final washing step, 57 ng/mL horseradish peroxidase (HRP)-conjugated streptavidin (Senova GmbH, Weimar, Germany), dissolved in HRP-Protector (Candor Bioscience GmbH, Wangen i. Allgäu, Germany) was added for 30 min at room temperature. Colorimetric analysis was facilitated by adding a substrate reagent (BioFX TMB Super Sensitive One Component HRP Microwell Substrate, Surmodics, Inc., Eden Prairie, MN, USA) for 15 min at room temperature. The colorimetric reaction was stopped by adding 2 N H_2_SO_4_ and absorbance read at 450 nm, corrected by absorbance at 570 nm, in a Microplate Reader (VICTOR X4 Multilabel Plate Reader, PerkinElmer, Waltham, MA, USA). As a control, soluble Hsp70 concentrations were measured using the DuoSet^®^ IC Human/Mouse/Rat Total Hsp70 ELISA (R&D Systems, Minneapolis, MN, USA) following the manufacturer’s protocol. A comHsp70 ELISA kit is currently under development by DRG Instruments GmbH, Marburg, Germany.

### 2.12. Isolation of Exosomes from EDTA Blood and Supernatants of Tumor Cells

Plasma isolated from EDTA blood (25 mL) was centrifuged (Thermoscientific Heraeus Megafuge 16R) with 4800× *g* for 30 min at 4 °C to remove cell debris. Then plasma was diluted 1:1 in ice-cold PBS (Life Technologies) and filtered through a 0.22 µm sterile filter (TPP) and ultracentrifuged (Sorvall discovery M120) at 150,000× *g* for 8 h. After one washing step in PBS and another ultracentrifugation for 4 h the pellet containing exosomes were resuspended in PBS and protein content was evaluated by the BCA-based protein assay. Exosomes were isolated from cell culture supernatants of 60–70% confluent tumor cell lines, as described above. The amount of Hsp70 derived from exosomes of the FBS in fresh medium was subtracted. Size and uniformity of the exosomes were characterized by dynamic light scattering on a Zetasizer NanoS instrument (Malvern Instruments, Malvern, UK) and by their protein content was determined by Western blot analysis using antibodies directed against β-actin (A228; Sigma-Aldrich), Hsp70 and Grp75 (SPS-825; StressMarq Biosciences Inc.).

### 2.13. Validation of the compHsp70 ELISA

To determine intra-assay precision, 36 different control sera were run twice on a 96-well ELISA plate. Inter-assay precision was assessed by running 39 serum samples in duplicates on three different 96-well ELISA plates. The concentration of each sample was determined, and the coefficient of variation (CV) was calculated using the ratio of standard deviation divided by the mean value. The Limit of Detection (LoD) was calculated in 42 blank samples and 42 samples with the lowest concentration of recombinant Hsp70 (1.56 ng/mL) according to the Clinical Laboratory Standards Institute (CLSI) guideline EP17-A, as described by Armbruster & Pry [46]. The Limit of Blank (LoB) was calculated as follows: LoB = mean_blank_ + 1.645 (SD_blank_) and the Limit of Detection (LoD) was calculated according to the following equation: LoD = LoB + 1.645 (SD_low concentration sample_). When 1.645 SD was used, no more than 5% of the values are less than the LoB. LoD is considered verified if it meets this criterion. Recombinant Hsp70 and Hsp70 in artificial lipid vesicles were measured in buffer and plasma of healthy donors. Recovery in plasma was assessed by spiking defined amounts of Hsp70 or Hsp70-containing vesicles in StabilZyme Select Stabilizer (Diarect GmbH, Freiburg i. Breisgau, Germany) into the plasma of healthy volunteers at a dilution of 1:5. The basal Hsp70 concentrations of the plasma into which the Hsp70 standards were spiked were subtracted from the values after spiking.

### 2.14. Statistical Tests

Data of all groups were compared using one-way analysis of variances (ANOVA) with Tukey test for multiple comparisons (RStudio). The pairwise comparison of groups was performed using a two-sided *t*-test (MS Excel), as recommended by our medical statistician.

## 3. Results

### 3.1. Epitope Mapping of the cmHsp70.1 and cmHsp70.2 Monoclonal Antibodies (mAbs)

Free Hsp70 in the circulation of tumor patients predominantly originates from dying cells, whereas exosomal Hsp70 is actively released by viable tumor cells. Currently available Hsp70 ELISA systems are unable to quantify the amount of exosomal Hsp70 in serum and plasma samples due to their inability to recognize an altered conformation of lipid-bound Hsp70. Consequently, we have established a novel compHsp70 sandwich ELISA, which is based on the cmHsp70.1 and cmHsp70.2 mAbs. Epitope mapping of two antibodies using SPOT analysis [40,44,45] revealed that the recognition sites of both antibodies are localized within the C-terminal substrate binding domain of Hsp70. The predominant linear sequences of the antibody epitopes of the cmHsp70.1 and cmHsp70.2 mAbs are N-L-L-G-R-F-E-L-S-G (aa 454–461) [40] and A-G-G-P-G-P-G-G-F-G (aa 614–623, as determined in the present study), respectively. A comparative inter-species analysis of the complete aa sequence of the major-stress inducible Hsp70 (HSPA1A) using the UniProt database in different species shows homologies of the human sequence with canine, rat, mouse and drosophila of 99%, 97% and 76%, respectively. The 8-mer epitope of the cmHsp70.1 mAb (aa 454–461; N-L-L-G-R-F-E-L) in the HSPA1A sequence is 100% identical in human, mouse, rat, dog, bovine, horse and pig, whereas a single aa exchange is present in the respective sequence of human Hsp70 (HSPA1A) and the highly homologous human Hsc70 (HSPA8) (aa 458; R to K). This aa exchange is also present in the HSPA1A epitope sequence of zebrafish. Since the sequence of feline HSPA1A is not available, human HSPA1A was compared to feline HSPA2, which shows two aa exchanges at positions 458 (R to K) and 460 (E to D). The inter-species similarities of the 10-mer sequence of the cmHsp70.2 mAb (aa 614–623; A-G-G-P-G-P-G-G-F-G) containing the antibody epitope was 100% conserved in human and pig, but there are 2 aa exchanges at positions 616 and 619 in human versus mouse and rat (G to A and P to A, respectively). The aa exchange at position 619 is also present in the relevant Hsp70 sequence of dog, bovine and horse. The 10-mer sequence of human HSPA1A differs in 3 positions to that of feline HSPA2 (aa 615, aa 620, aa 623) and in 5 positions in HSPA1A of human versus zebrafish (aa 614, aa 617, aa 618, aa 619, aa 622) (Table 1). All documented exchanges in the different species are conservative and nonpolar [47,48]. The epitope similarities of the cmHsp70.1 and Hsp70.2 mAbs indicates that, in addition to humans, the compHsp70 ELISA system is likely to be capable of measuring free and exosomal Hsp70 in the blood of different mammalian species, including dog, bovine, horse and pig.

### 3.2. Affinities of cmHsp70.1 and cmHsp70.2 mAbs to Recombinant Hsp70 Protein

The K_D_ values of the cmHsp70.1 and cmHsp70.2 mAbs to recombinant Hsp70 protein, as determined by Microscale Thermophoresis (MST) measurements, were 0.42 nM and 0.44 nM, respectively (Figure 1a,b).

### 3.3. Binding Characteristics and Specificities of Hsp70 Antibodies to Recombinant Hsp70 Protein, Tumor Cell Lysates and Lipid-Bound Hsp70 Protein

The binding characteristics of the cmHsp70.1 and cmHsp70.2 mAbs with respect to their capacity to detect free and lipid-bound Hsp70 were compared with those of two commercially available Hsp70 control antibodies (ctrl Hsp70A, ctrl Hsp70B) using Western and dot blot analysis of HSP proteins and flow cytometric analysis of viable, membrane Hsp70-positive tumor cells. As shown in Figure 2a, all tested antibodies detected recombinant Hsp70 protein (50 ng) and Hsp70 in tumor cell lysates (50 µg), as determined by Western blot analysis. In contrast to the other antibodies ctrl Hsp70B antibody detects in addition to Hsp70 (72kDa) also Hsc70 (73 kDa) as shown by a faint band at 73 kDa. To determine the capacity of the different Hsp70 antibodies to detect membrane-bound or liposomal Hsp70, flow cytometric analyses were performed using viable A549 (adenocarcinoma alveolar basal epithelial cells) and HCT116 (colon carcinoma) cells. Tumor cells were incubated with equal amounts of FITC labeled cmHsp70.1, cmHsp70.2, ctrl Hsp70A and ctrl Hsp70B antibodies; isotype-matched control antibodies served as negative controls. A representative FACS analysis is shown in Figure 2b. A positive cell surface staining using the antibodies cmHsp70.1 and cmHsp70.2 was detected in viable A549 (81.7 ± 6.6% vs. 80.0 ± 6.9%; *n* = 3) and HCT116 cells (90.5 ± 3.7% vs. 93.1 ± 4.5; *n* = 3). In contrast, the ctrl Hsp70A and ctrl Hsp70B antibodies failed to detect membrane-bound Hsp70 on the cell surface of either tumor cell line. Peripheral blood lymphocytes (PBL) of healthy human donors (*n* = 5) exhibited no Hsp70 surface staining either with cmHsp70.1 (2.1 ± 1.4%) or cmHsp70.2 (1.5 ± 1.8%) mAbs. A representative example of a cell surface staining of viable PBL with cmHsp70.1 and cmHsp70.2 is illustrated in Figure 2b.

Specificity of the antibodies was tested by dot blot analysis using bovine serum albumin (BSA), Hsp27, Hsp60, Hsp70 and Hsc70 proteins at different concentrations ranging from 100 ng to 0.1 ng against cmHsp70.1/2 and ctrl Hsp70A/B antibodies. In line with the data of the Western blot, the dot blot analysis demonstrates that cmHsp70.1/2 and ctrl Hsp70A antibodies specifically detect Hsp70, and do not cross-react with BSA or other HSPs such as Hsp27, Hsp60 or the highly homologous Hsc70. In contrast, the ctrl Hsp70B antibody reacts with both Hsc70 and Hsp70 at the two highest protein concentrations (100 ng, 0.1 ng) (Figure 2c).

### 3.4. Calibration Curve and Inter- and Intra-Assay Precision of the compHsp70 ELISA Using cmHsp70.1 and cmHsp70.2 mAbs

Since the cmHsp70.1 and cmHsp70.2 mAbs are able to bind both, free and membrane-bound Hsp70, these antibodies were used to develop the compHsp70 sandwich ELISA. Comparative analysis revealed that cmHsp70.2 mAb qualifies best as the coating antibody and the cmHsp70.1 mAb as the detection antibody. An eight-point standard calibration curve (0 to 100 ng/mL) was established using purified Hsp70 protein. A representative calibration curve (0 to 100 ng/mL), its regression equation, and the coefficient of determination (R^2^ value) are illustrated in Figure 3. In 42 independent ELISA experiments, the standard curve showed a high reproducibility with R^2^ values ranging between 0.974 and 1.000. The highest concentration of the standard (100 ng/mL) typically yielded a mean OD of 2.26 ± 0.33 arbitrary units (a.u.). The linearity of the compHsp70 ELISA remains stable within a concentration range of 1.56–25.00 ng/mL.

The intra-assay precision coefficients of variation (CV) ranged between 0.02% and 12.50%, as determined by analyzing 36 different serum samples in duplicate. The inter-assay precision coefficient (CV) varied between 0.38% and 7.34%, as determined by analyzing 39 serum samples in duplicate in three independent ELISA experiments (Table 2).

The recovery rates were determined after spiking different concentrations of recombinant Hsp70 protein into buffer and plasma samples of healthy volunteers. After spiking 2.5 ng/mL Hsp70 into buffer, the recovery rate by the compHsp70 ELISA was 105 ± 1%. After spiking 15 ng/mL Hsp70 into plasma samples, the recovery rate was 92 ± 9%. In the latter setting, the intrinsic Hsp70 content of the plasma sample was subtracted from the measured value. To further characterize the compHsp70 ELISA, the Limit of Detection (LoD), as calculated by using the equation LoD = LoB (Limit of Blank) + 1.645 (SD_low concentration sample_), was determined to be 4.37 ng/mL (Table 2).

### 3.5. Recovery of Spiked Liposomal Hsp70 Using cmHsp70.1 and Ctrl Hsp70A mAbs as Detection Antibodies

To evaluate the capacity of cmHsp70.1 and ctrl Hsp70A mAbs to detect lipid-bound Hsp70 in the compHsp70 ELISA, artificially manufactured liposomes loaded with Hsp70 were prepared as artificial extracellular vesicle surrogates. Hsp70-loaded artificial liposomes (liposomal Hsp70: 500 ng/mL) were spiked into the serum of healthy volunteers and cmHsp70.2 mAb was used as a coating antibody in the ELISA. The recovery of liposomal Hsp70 using cmHsp70.1 and ctrl HSP70A mAbs as detection antibodies was 421.8 ± 34.6 ng/mL (black bar) and 113.2 ± 6.6 ng/mL (grey bar), respectively (Figure 4a), which corresponds to recovery rates of 84.4 ± 9.9% (black bar) and 22.7 ± 1.3% (grey bar), respectively (Figure 4b). These data indicate that the recovery of lipid-bound Hsp70 by the cmHsp70.1 mAb was nearly four-fold higher than that of the ctrl Hsp70A mAb when used as a detection antibody. The ctrl Hsp70B mAb was unable to detect any liposomal Hsp70 when used as a detection antibody in an identical experimental setting. The ctrl Hsp70A and ctrl Hsp70B mAbs also did not function as effective coating antibodies in the compHsp70 ELISA.

### 3.6. Impact of Interference Factors on Hsp70 Levels in the Blood Determined by the compHsp70 ELISA

To investigate the robustness of the data obtained with the compHsp70 ELISA using cmHsp70.2 mAb for coating and cmHsp70.1 mAb for detecting, serum and plasma samples from 13 healthy volunteers were collected. As shown in Figure 5a, Hsp70 concentrations measured in the plasma and serum by the compHsp70 ELISA were not significantly different. These data indicate that the compHsp70 ELISA can measure free and vesicular Hsp70 in both plasma and serum samples. To determine the influence of food intake on the detection of Hsp70 in the blood, serum samples were taken from 17 healthy individuals before and 2 h after an intake of a high fat diet. In all cases, serum Hsp70 concentrations before and after food intake were the same (Figure 5b). Since the age of the donors might have an impact on the Hsp70 concentrations in the circulation, plasma samples of 108 volunteers in different age groups ranging from 21–77 years (Table 3) were analyzed. As shown in Figure 5c, there was no significant correlation between plasma Hsp70 concentrations and the age of the donors, as determined using the Pearson correlation test (R^2^ = 0.0781).

### 3.7. Comparative Analysis of Hsp70 Concentrations in the Blood and Exosomes of Cancer Patients and Healthy Donors

In a first clinical evaluation, serum Hsp70 concentrations in patients with non-small cell lung carcinoma (NSCLC; *n* = 166) and high grade gliomas (HGG; *n* = 34; 26 primary, 8 relapse) were determined using the compHsp70 ELISA and compared to those in healthy volunteers (*n* = 108). The assay was compared with samples of the same matrici. Serum samples of healthy donors were diluted 1:5 in dilution buffer, whereas serum samples of tumor patients with high Hsp70 levels were diluted 1:20. The mean serum Hsp70 concentrations in patients with HGG (91.8 ± 21.3 ng/mL) and NSCLC (332.2 ± 37.9 ng/mL) were significantly higher than those in healthy volunteers (35.1 ± 4.0 ng/mL; Figure 6a). Receiver Operating Characteristic (ROC) curve analysis compared serum Hsp70 concentrations of healthy individuals with those of NSCLC and HGG patients (Figure 6b). The Area Under the Curve (AUC), the CI 95% value, the sensitivity and the specificity for a cut-off value was 114 ng/mL for NSCLC and 6 ng/mL for HGG patients, as determined by calculating the Youden-Index (Table 3). For validation, the assay was performed by at least five skilled operators, on different days and with at least three different reagent lots.

**Table 3 cancers-13-03706-t003:** Characteristics and mean Hsp70 concentrations of healthy donors and patients with NSCLC and HGG. Used abbreviations in the Table: Area Under the Curve (AUC); Male (M); Female (F); Confidence Interval (CI); high grade glioma (HGG); Non-small cell lung cancer (NSCLC); Receiver Operating Characteristic (ROC); Standard Deviation (SD); Standard Error of the Mean (SEM). A two-sided *t*-test was used.

	Parameters	Healthy Donors	NSCLC	HGG
Number (*n*)		108	166	34
Gender (m/f)		51/57	109/51	28/6
Age	Mean	43	67	59
	Range	21–77	41–91	21–84
	SD	16	10	14
	Median	44	67	60
compHsp70 ELISA	Mean Hsp70 (ng/mL)	35.06	332.19	91.82
	SEM	3.99	37.90	21.28
ROC	AUC (CI 95%)		0.88	0.62
	*p*-value		<0.00001	0.03
	Sensitivity (%)		68	91
	Specificity (%)		94	33
	Threshold (ng/mL)		114	6

The lower specificity in HGG vs. NSCLC could be explained by lower values of Hsp70 in HGG caused by the blood brain barrier, which limits the transport of vesicular Hsp70 into the circulation.

Patients with NSCLC were classified according to their histology (squamous cell carcinoma, adeno carcinoma) and tumor stage (stage I to IV). Mean Hsp70 concentrations were given in tumor subgroups containing more than 4 patients (Table 4).

Compared to healthy individuals (35 ± 3.99 ng/mL; *n* = 108), serum Hsp70 levels were significantly higher in patients with stage III and IV squamous cell carcinoma of the lung (234.4 ± 29.2 ng/mL and 321.2 ± 68.8 ng/mL, respectively) as well as in patients with stage III and IV adeno lung carcinoma (260.3 ± 75.3 ng/mL and 561.3 ± 173.4 ng/mL, respectively). For both tumor entities, Hsp70 concentrations were higher in patients with stage IV disease than those with stage III disease (* *p* < 0.05). Furthermore, Hsp70 concentrations were higher in patients with adeno than squamous cell carcinoma histology, although these differences did not reach statistical significance (Figure 6c).

In addition to measuring circulating Hsp70 concentrations in tumor patients at diagnosis, Hsp70 levels were determined in responding patients with NSCLC before, during and after completion of radiotherapy. Blood samples were collected from patients before radiotherapy, during radiotherapy (after approximately 20 Gy; range 18–22.5 Gy) and directly after completion of radiotherapy (60–70 Gy). Hsp70 concentrations before radiotherapy (494.1 ± 72.2 ng/mL; *n* = 80), during radiotherapy (310.5 ± 36.8 ng/mL; *n* = 58) and after completion of radiotherapy (380.0 ± 51.8 ng/mL; *n* = 56) were significantly higher than those of healthy individuals (35.1 ± 3.99 ng/mL; *n* = 108) when measured using the compHsp70 ELISA (Figure 6d). After receiving a radiation dose of approximately 20 Gy (range 18–22.5 Gy) Hsp70 levels dropped significantly from 494.1 ± 72.2 to 310.5 ± 36.8 ng/mL. After completion of radiotherapy (60–70 Gy) the Hsp70 concentration was 380.0 ± 51.8 ng/mL. Hsp70 values measured with a control Hsp70 ELISA that only detects free Hsp70 were significantly lower and did not differ in the course of therapy (NSCLC before RT: 4.5 ± 1.3 ng/mL; *n* = 80; during RT: 3.5 ± 1.0 ng/mL; *n* = 58; after RT: 3.8 ± 1.2 ng/mL; *n* = 56).

The Hsp70 content was also assessed comparatively in exosomes isolated from the EDTA-blood of a healthy human donor, a tumor patient with squamous cell carcinoma (Figure 6e) and the supernatant of a tumor cell line grown at a 60–70% confluency (Figure 6f). Size and purity of plasma-derived exosomes was demonstrated by size distribution intensity measurements (Figure 6e, upper). Dynamic light scattering measurements revealed comparable patterns in the size distribution of exosomes of a healthy human donor and a tumor patient. The size of the main particle fraction was determined as 88 ± 48 nm with an intensity of 86.5% of the total particle content in the healthy donor, which was accompanied by a second peak at 14 ± 4 nm at an intensity of 13.5%. For the tumor patient, the size distribution revealed peaks at 86 ± 38 nm (intensity 84%) and a second peak at 12 ± 3 nm (intensity of 14%) (Figure 6e, upper). The exosomal Hsp70 levels measured with the compHsp70 ELISA were more than five-fold higher in the tumor patient compared to the healthy human donor (Figure 6e, lower). The ctrl Hsp70 ELISA neither detected any significant amounts of exosomal Hsp70 in the plasma of the tumor patient nor of the healthy human donor (Figure 6e). Similar results were obtained with exosomes derived from the supernatant of a tumor cell line. The compHsp70 ELISA, but not the control ELISA was able to detect exosomal Hsp70 derived from the supernatant of membrane Hsp70-positive tumor cells grown at a 60–70% confluency. As expected, the exosomal fraction contained only cytosolic proteins (β-actin, Hsp70 determined by ELISA), whereas the corresponding tumor cell lysate contained cytosolic (β-actin) and ER-residing (Grp75) proteins.

## 4. Discussion

Liquid biomarkers detecting tumor-derived factors (e.g., proteins, small extracellular vesicles, circulating tumor DNA, circulating tumor cells or circulating microRNAs) have the potential to improve tumor detection, diagnosis, prognosis, and the prediction of therapeutic responses [49,50]. Another advantage of liquid biopsies is the possibility of a repeated sample collection using a minimally invasive method. Circulating biomarkers could provide clinically relevant information about the pathophysiology of the tumor, response to radio- and/or chemotherapies, and insight into the risk of developing metastatic disease and/or relapse, in *real-time.* Therefore, liquid biomarkers might enable a more accurate patient stratification and better-tailored therapy decisions. Presently, only a few circulating protein-based biomarkers with relatively low specificities (e.g., cytokeratin 19 fragment (CYFRA 21-1, specificity 76%), carcinoma embryonic antigen (CEA, specificity 52%), carbohydrate antigen 125 (CA125, specificity 52%), carbohydrate antigen 153 (CA153), carbohydrate antigen 199 (CA199) and neuron-specific enolase (NSE, specificity 22%)) have been incorporated into clinical practice for NSCLC [51,52,53,54,55,56]. However, as single markers their prognostic value is controversially discussed, and their use is often limited to certain tumor subtypes [57].

This study has established a novel compHsp70 sandwich ELISA, which is based on two monoclonal antibodies that can be produced in unlimited amounts with a high quality and a reproducible specificity and sensitivity. The Hsp70 values derived with a previously established Hsp70 ELISA system based on only one monoclonal antibody and a polyclonal rabbit antiserum were generally lower and due to variations in the polyclonal Hsp70 rabbit antiserum the values were not stable. Both monoclonal antibodies cmHsp70.1 and cmHsp70.2 are able to detect free and vesicular Hsp70. Free Hsp70 predominantly origins from dying tumor cells, whereas Hsp70 in small extracellular vesicles is actively released by a large variety of highly aggressive, membrane Hsp70-positive viable tumor cells [31]. Vesicular Hsp70 might therefore be a valuable biomarker for determining the viable tumor mass at diagnosis and during or after therapy in different tumor entities. Most commercially available Hsp70 ELISA systems are validated only for the detection of free Hsp70 in aqueous solutions. The compHsp70 ELISA allows the quantification of tumor-derived Hsp70 in serum and plasma with high precision and linearity in a clinically relevant concentration range. The specificity of the ELISA is documented by the fact that both compHsp70 ELISA antibodies only detect the major-stress inducible Hsp70 (HSPA1A), but do not cross-react with other HSP proteins such as Hsp27, Hsp60 or its highly homologous constitutively expressed family member, Hsc70 (HSPA8). Due to a high inter-species homology, the epitopes of the two Hsp70 ELISA antibodies are conserved in different species and therefore, the compHsp70 ELISA might qualify to measure circulating Hsp70 in different species.

An impact of different factors such as lifestyle, age and gender on levels measured by the comHsp70 ELISA was excluded as the analysis of serum and plasma samples of 108 healthy individuals in different age groups before and after intake of a high fat diet did not reveal any significant correlations.

Compared to healthy individuals, patients with late-stage squamous cell and adeno carcinoma of the lung and gliomas revealed significantly higher Hsp70 levels in the circulation at diagnosis. Since a major part (approximately 100-fold more than free Hsp70) of extracellular Hsp70 originates from small extracellular vesicles and since the compHsp70 ELISA was shown to reliably detect spiked liposomal Hsp70 with a high sensitivity in the blood, it is very likely that elevated Hsp70 levels in the circulation of patients reflect the viable tumor mass. These data are in line with other studies reporting on elevated levels of chaperone-containing vesicles in tumor patients. Extracellular vesicles of unstressed and stressed tumors fulfill many different tasks including mediating cytotoxic anti-cancer immune responses [58,59], increasing tumor cell motility and growth, eliciting tumor-specific immunity, promoting angiogenesis [60,61], inducing Th17 cells, eliciting NK cell mediated immunity [62,63] and transferring therapy resistance [64]. In our study, significantly higher Hsp70 levels were detected in patients with squamous cell carcinoma of the lung (*p* < 0.001) and glioblastoma (*p* < 0.03) compared to healthy individuals. HGG Hsp70 was also detectable in the liquor of patients, but at low levels. Moreover, the Hsp70 levels increase with higher tumor stages and thereby might serve as a biomarker for risk assessment. Subsequent receiver operating characteristic (ROC) curve analysis allowed a discrimination of Hsp70 serum levels in lung and brain cancer patients and healthy individuals. Given that, in addition to lung and brain tumors, membrane-Hsp70 is also present on the plasma membrane of a large variety of different other tumor types from which it is released into the extracellular milieu, it is reasonable to assume that extracellular Hsp70 might serve as a universal tumor biomarker in a broad range of cancer entities.

As previously reported [34,65], blood Hsp70 levels correlate with the intracellular Hsp70 levels and match the membrane-Hsp70 status of the tumor cells from which they originate. In this study, we observed a significant decrease in the extracellular Hsp70 levels in patients with lung carcinoma during (after approximately 20 Gy) and after completion of radiotherapy (60–70 Gy). In contrast, no significant drop in Hsp70 values was detected after radiotherapy when a commercial Hsp70 ELISA was used, and the values measured with the novel compHsp70 ELISA were approximately more than 100-fold higher. The decrease in circulating Hsp70 in the peripheral blood of responding tumor patients likely indicates a reduction in viable tumor mass in response to ionizing radiation. The minor, but not significant, increase of Hsp70 after completion of radiotherapy most likely attributes to an increased presence of free Hsp70 in the circulation derived from dying tumor cells and radiation-induced inflammation. To demonstrate that the circulating Hsp70 in tumor patients, which is detected by the compHsp70 ELISA originates from exosomes we isolated extracellular vesicles from a tumor patient, the supernatant of a membrane Hsp70-positive tumor cell line and a healthy control donor. Size distribution and protein content characterized the isolated extracellular vesicles as exosomes. A comparison of the Hsp70 content in isolated exosomes revealed more than five-fold higher Hsp70 levels in plasma-derived exosomes of a tumor patient compared to that of a healthy individual. Hsp70 can be measured with the compHsp70 ELISA also in the exosomal fraction derived from the cell culture medium of the membrane Hsp70-positive tumor cell line. As a control, no Hsp70 is detected in any of the exosomal fractions with a control Hsp70 ELISA, which is known to react only with free Hsp70.

In line with our findings, a prospective clinical study including patients with solid tumors (breast and NSCLC) has demonstrated exosomal Hsp70 levels to inversely correlate with therapeutic response [66]. In this study, a protocol that allows the isolation of small extracellular vesicles from plasma samples of patients for a molecular characterization of exosomal proteins was established. However, the complexity and time-consuming nature of isolating these vesicles limits its clinical application. The simultaneous evaluation of free and exosomal Hsp70 in serum and plasma of tumor patients using the compHsp70 ELISA might provide valuable and actionable clinical information for predicting therapeutic outcome in the future.

Herein, we demonstrate that exosomal Hsp70 levels might qualify for estimating the clinical response to radiotherapy in lung carcinomas with adeno and squamous cell histologies, since patients who responded to radiotherapy showed a drop in exosomal Hsp70 levels. In previous studies we could show that exosomal Hsp70 values correlate with the gross tumor volume [65] and a combined assessment of Hsp70 and the hypoxia-related marker osteopontin is superior in predicting clinical responses in NSCLC patients [67]. Therefore, monitoring the dynamics of Hsp70 together with other tumor biomarkers in circulation during therapy might allow a faster therapy adaptation and thereby contribute to a better outcome. Interference with isocitrate dehydrogenase (IDH) provides a promising strategy in the treatment of HGG. Preclinical studies demonstrated that inhibitors of mutant IDH-1 and IDH-2 prevent the accumulation of the oncometabolite d-2-hydroxyglutarate (2-HG) in HGG [68]. Vorasidenib (AG-881) inhibits the production of 2-HG in glioma tissue by >97% in an orthotopic glioma mouse model [69]. In a case report, treatment with ivosidenib showed an improved seizure control and radiographic stable disease in a glioblastoma patient for more than 4 years [70]. An IDH1 (R132H)-specific peptide vaccine (IDH-1-vac) inducing a tumor-specific T helper cell response demonstrated efficacy in IDH-1(R132H)^+^ tumors in preclinical models and a first-in-man clinical trial [71].

Since extracellular Hsp70 serves as a surrogate for the Hsp70 membrane status of the tumor, theranostic approaches [72] targeting membrane Hsp70-positive tumors using antibodies, immune effector cells [73] and other approaches based on targeting membrane-Hsp70 could be based on the data derived with the compHsp70 ELISA. Despite promising results, longitudinal follow-up studies with different patient cohorts are necessary to fully evaluate and validate the clinical value of using the compHsp70 ELISA for monitoring tumor recurrence and metastasis and for the correlation of exosomal Hsp70 levels with progression-free survival (PFS) and overall survival (OS) in cancer patients. The prognostic value can be further increased by combining the data on exosomal Hsp70 with that of other biomarkers in liquid biopsies such as circulating free tumor cell DNA and/or mircoRNA that have been found to be informative for prediction of responses to EFGR inhibitor therapies in NSCLC. However, a major limitation of these biomarkers are false negative results, which are caused by different factors including the low signal-to-noise ratio and the short half-life (<1.5 h) of the circulating DNA/RNA [74]. When soluble Hsp70 is used as a biomarker tumor progression can be assessed because Hsp70 is predominantly released by viable tumor cells, and therefore Hsp70 might serve as biomarker for the viable tumor mass [65]. Furthermore, the half-life of the circulation of Hsp70 (particularly under stress conditions) is much longer (in the range of 7 h) than that of ctDNA (<1.5 h) [75]. In this regard, soluble Hsp70 provides a relatively stable tumor biomarker in the blood, which enables the assessment of the viable tumor mass, and therefore might be able to predict tumor responses.

This hypothesis is in line with a report of Tomita et al., which has shown that the combined monitoring of CYFRA 21-1 and CEA, as relevant biomarkers in patients with NSCLC, improves prognostic relevance [76].

## 5. Conclusions

The novel compHsp70 ELISA presented herein provides a reliable and robust tool to quantify free and vesicular Hsp70 in the serum and plasma of cancer patients, levels of which might reflect the presence and risk characteristics of tumors, their membrane-Hsp70 status and therapeutic response.

## Figures and Tables

**Figure 1 cancers-13-03706-f001:**
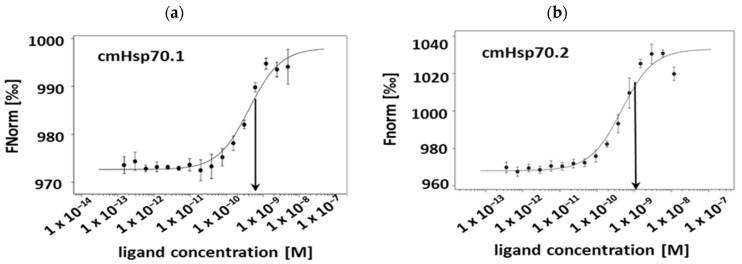
Binding affinity of the cmHsp70.1 and cmHsp70.2 mAbs. The binding affinity of cmHsp70.1 (K_D_ 0.42 nM) (**a**) and cmHsp70.2 (K_D_ 0.44 nM) (**b**) to Hsp70 protein was measured using the Microscale Thermophoresis technique. The *X*-axis represents the concentration of the Hsp70 protein (M) and the Fnorm (‰) is displayed on the *Y*-axis. Arrows indicate the determined K_D_ values.

**Figure 2 cancers-13-03706-f002:**
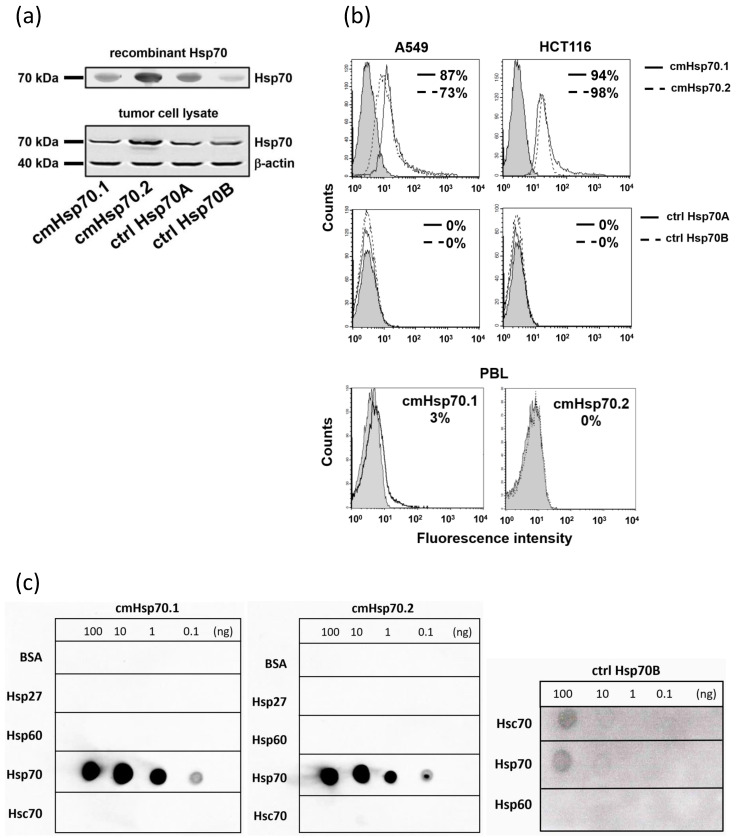
Western blot of recombinant Hsp70 protein and tumor cell lysates, flow cytometric analysis of membrane-bound Hsp70 on viable tumor cells, and dot blot analysis using BSA, recombinant Hsp27, Hsp60, Hsp70, Hsc70 as target proteins. (**a**) Hsp70 protein and lysates of tumor cells were subjected to an SDS gel and blots were stained with cmHsp70.1, cmHsp70.2, ctrl Hsp70A and ctrl Hsp70B antibodies. Representative immunoblots of recombinant Hsp70 (upper, 72 kDa) and tumor cell lysates (lower) representing Hsp70 (72 kDa), Hsc70 (73 kDa) and β-actin (40 kDa) as a loading control are shown. Molecular weight markers are indicating 70 kDa and 40 kDa. Detailed information about Western blot can be found at Appendix A. (**b**) Detection of membrane-bound Hsp70 on A549, HCT116 cell lines by flow cytometry using the cmHsp70.1, cmHsp70.2, ctrl Hsp70A and ctrl Hsp70B antibodies (white histograms). Staining with the respective isotype-matched control antibodies is represented by grey histograms. As a control, PBL of healthy donors were stained with cmHsp70.1 and cmHsp70.2 mAb. The numbers in the histograms indicate the proportion of Hsp70 positively stained cells. (**c**) Dot blot analysis of BSA, recombinant Hsp27, Hsp60, Hsp70 and Hsc70 (100 ng, 10 ng, 1 ng, 0.1 ng) using cmHsp70.1, cmHsp70.2 and ctrl Hsp70B antibodies.

**Figure 3 cancers-13-03706-f003:**
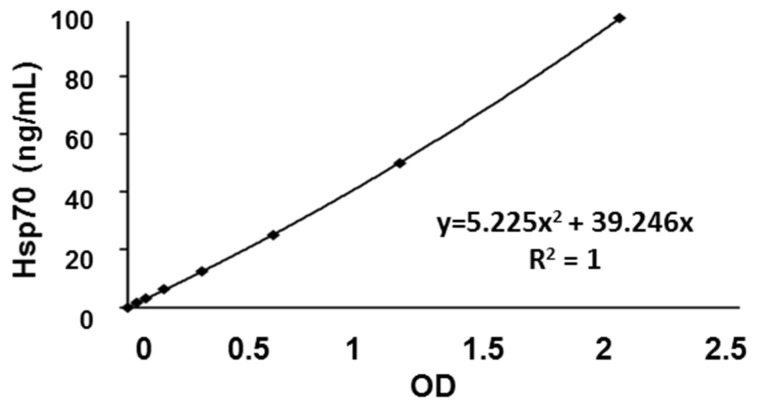
Representative eight-point calibration curve for measuring samples of healthy individuals. The *X*-axis and *Y*-axis represent the OD (a.u.) and the Hsp70 protein concentration (ng/mL), respectively.

**Figure 4 cancers-13-03706-f004:**
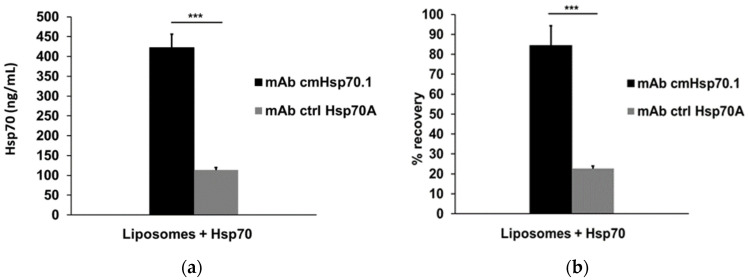
Detection and recovery of liposomal Hsp70 in serum. Quantification (**a**) and recovery (**b**) of lipid-bound Hsp70 in artificial lipid vesicles (liposomal Hsp70: 500 ng/mL), as determined by the sandwich Hsp70 ELISA using the cmHsp70.2 mAb as a coating antibody, and cmHsp70.1 and ctrl Hsp70A mAbs as detection antibodies. The data represent mean values of three independent experiments, *** *p* < 0.001, a two-sided *t*-test was used.

**Figure 5 cancers-13-03706-f005:**
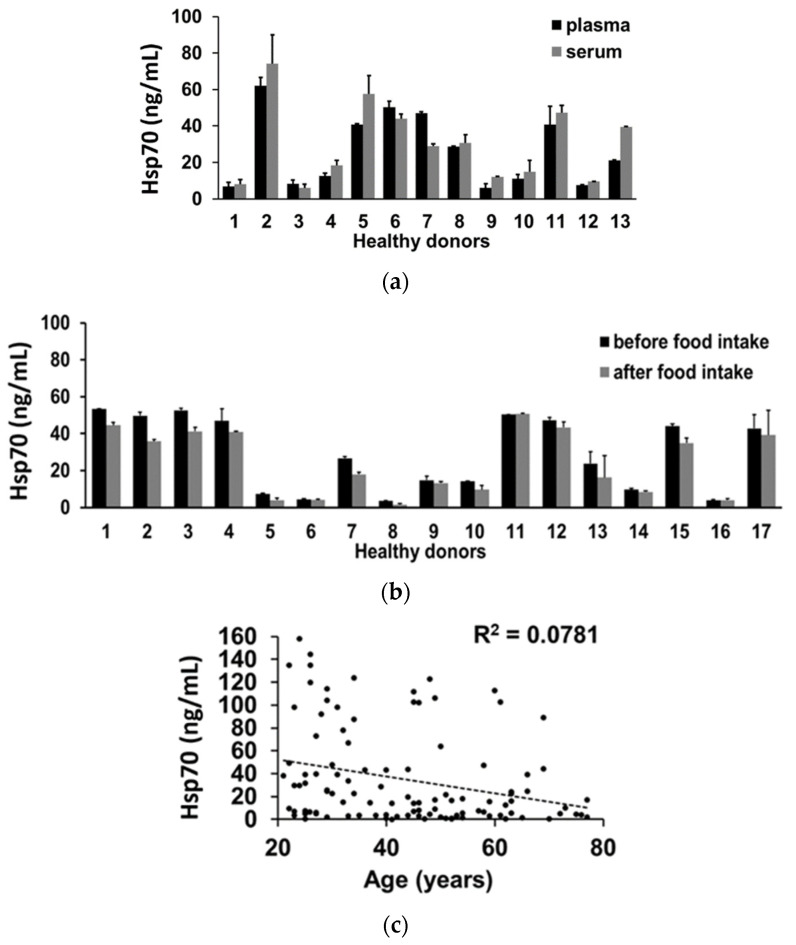
Interference factors potentially influencing the Hsp70 concentrations measured by the compHsp70 ELISA. (**a**) Comparison of the Hsp70 concentrations in plasma and serum. Plasma (black bars) and serum (grey bars) from 13 healthy individuals were taken at the same time. (**b**) Hsp70 concentrations in the serum of 17 healthy individuals taken before (black bars) and 2 h after a high fat diet (grey bars). (**c**) Hsp70 concentration of 108 healthy volunteers in different age groups ranging from 21–77 years. Characteristics of the healthy donors are summarized in Table 3. A Pearson correlation test was performed.

**Figure 6 cancers-13-03706-f006:**
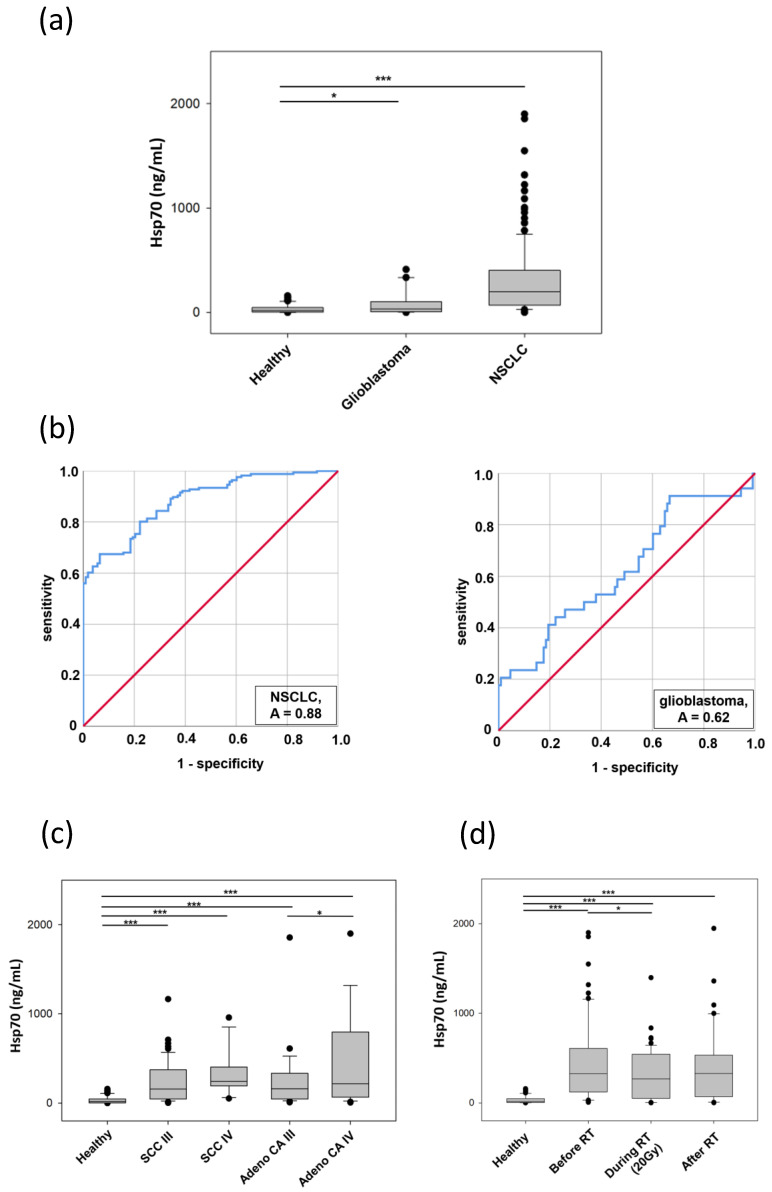
Serum and exosomal Hsp70 concentrations in healthy donors and tumor patients. (**a**) Serum Hsp70 concentrations in healthy individuals (*n* = 108), patients with NSCLC (*n* = 166) and HGG (*n* = 34), as measured using the compHsp70 ELISA. Significantly higher Hsp70 levels were found in both tumor patient cohorts compared to healthy controls (* *p* < 0.05, *** *p* < 0.001, Mann-Whitney U Test). (**b**) ROC curve analysis using the data shown in (**a**). (**c**) Serum Hsp70 concentrations in squamous cell carcinoma patients stage III (234.4 ± 29.2 ng/mL; *n* = 59) and IV (321.2 ± 68.8 ng/mL; *n* = 12) as well as adeno carcinoma patients stage III (260.3 ± 75.3 ng/mL; *n* = 24) and IV (561.3 ± 173.4 ng/mL; *n* = 29) compared to Hsp70 levels of healthy individuals (35.1 ± 3.99 ng/mL; *n* = 108) (*p* < 0.001, two-sided *t*-test). Differences in serum Hsp70 concentrations in adeno carcinoma stage III and IV were significantly different (* *p* < 0.05; ANOVA Tukey test) (**d**) Serum Hsp70 concentrations in patients before radiotherapy (Before RT; 494.1 ± 72.2 ng/mL; *n* = 80), during radiotherapy (During RT; 310.5 ± 36.8 ng/mL; after approximately 20 Gy; *n* = 58) and after radiotherapy (After RT; 380.0 ± 51.8 ng/mL; *n* = 56), compared to healthy individuals (Healthy; 35.1 ± 4.0 ng/mL; *n* = 108), as measured by the compHsp70 ELISA (*** *p* < 0.001, *t*-test). Following a dose of approximately 20 Gy (range 18–22.5 Gy), the Hsp70 levels dropped significantly from 494.1 ± 72.2 to 310.5 ± 36.8 ng/mL (* *p* < 0.05, *t*-test). For pairwise comparisons, two-sided t-test was applied, for comparison of all groups, ANOVA Tukey test was used. (**e**) Hsp70 content in exosomes isolated from the plasma of a healthy control donor, a tumor patient, as determined by the compHsp70 ELISA and a control (ctrl) Hsp70 ELISA that only detects free Hsp70. Size and purity of the plasma-derived exosomes from the plasma was determined by dynamic light scattering. (**f**) Hsp70 content in exosomes isolated from the supernatant of tumor cells, as determined by the compHsp70 ELISA and a control (ctrl) Hsp70 ELISA and protein content in tumor cell lysates and exosomes, as determined by Western blot analysis. Detailed information about Western blot can be found at Appendix A.

**Table 1 cancers-13-03706-t001:** Comparison of Hsc70 (HSPA8) and Hsp70 (HSPA1A) and inter-species comparison of the 8-mer (aa 454–461) and 10-mer (aa 614–623) sequences of Hsp70 (HSPA1A) in humans and different other species based on the UniProt database. The given HSP70 sequences contain the epitopes of the cmHsp70.1 and cmHsp70.2 mAbs. (*) For feline Hsp70, only the aa sequence of HSPA2 is available. Amino acid (aa) exchanges are marked in grey.

Species	UniProt ID HSP70 Names	Total aa	Epitope of cmHsp70.1 (8-mer) aa Sequence	Epitope of cmHsp70.2 (10-mer) aa Sequence
Human	P11142 HSPA8, Hsc70	646	N L L G K F E L 454–461	A G G M P G G M P G 614–623
Human	P0DMV8 HSPA1A, Hsp70	641	N L L G R F E L 454–461	A G G P G P G G F G 614–623
Mouse	Q61696 HSPA1A, Hsp70	641	N L L G R F E L 454–461	A G A P G A G G F G 614–623
Rat	P0DMW0 HSPA1A, Hsp70	641	N L L G R F E L 454–461	A G A P G A G G F G 614–623
Dog	Q7YQC6 HSPA1A, Hsp70	641	N L L G R F E L 454–461	A G G P G A G G F G 614–623
Bovine	Q27975 HSPA1A, Hsp70	641	N L L G R F E L 454–461	A G G P G A G G F G 614–623
Horse	F7DW69 HSPA1A, Hsp70	641	N L L G R F E L 454–461	A G G P G A G G F G 614–623
Pig	P34930 HSPA1A, Hsp70	641	N L L G R F E L 454–461	A G G P G P G G F G 614–623
Cat	M3W8G1 HSPA2 *, Hsp70.2	639	N L L G K F D L 457–463	Q G G P G G G G S G 615–624
Zebrafish	B0S610 HSPA1A, Hsp70	643	N L L G K F E L 456–463	Q G G M P A G G C G 614–623

**Table 2 cancers-13-03706-t002:** Assay performance of the compHsp70 ELISA.

Parameters	Performance	Mean Values
Linear range (ng/mL)	1.56–25.00	
Intra-assay precision (% CV)	0.02–12.50	3.82%
Inter-assay precision (% CV)	0.38–7.34	3.64%
Recovery of Hsp70 (%) buffer (2.5 ng)/plasma (15 ng)	105 ± 1/92 ± 9	
Limit of Detection (LoD, ng/mL)	4.37	

**Table 4 cancers-13-03706-t004:** Histology, tumor stage and mean Hsp70 values of squamous cell and adeno non-small cell lung cancer (NSCLC) patients. Data uncertainty is given as Standard Error of the Mean (SEM). An asterisk marks statistically significant differences between NSCLC stage III and IV, as determined by the one-way analysis of variance (ANOVA) with the Tukey test for multiple comparisons in RStudio; * *p* < 0.05.

Histology	Stage	Number of Cases (*n*)	compHsp70 ELISA Hsp70 (ng/mL)
Squamous cell carcinoma	I	2	
Squamous cell carcinoma	II	4	
Squamous cell carcinoma	III	59	234.4 ± 29.2
Squamous cell carcinoma	IV	12	321.2 ± 68.8
Squamous cell carcinoma	Not specified	5	249.9 ± 64.4
Adeno carcinoma	I	1	
Adeno carcinoma	II	1	
Adeno carcinoma	III	24	260.3 ± 75.3
Adeno carcinoma	IV	29	561.3 ± 173.4 *
Adeno carcinoma	Not specified	2	
Not otherwise specified	Not specified	27	336.1 ± 65.0

## Data Availability

The data presented in this study are available on request from the corresponding author. The data are not publicly available due to IP restrictions.

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
