# Peer review of "Hsp70 in Liquid Biopsies—A Tumor-Specific Biomarker for Detection and Response Monitoring in Cancer"

_cancers, 2021, doi:10.3390/cancers13153706_

Round 1

Reviewer 1 Report

The authors of the manuscript ‘ Exosomal Hsp70 in liquid biopsies – a biomarker for prediction and response monitoring in cancer’ describe an ELISA-based method to detect cancer based on exosomal Hsp70. They describe a novel set of antibodies that are suggested to have superior performance over commercially available ones once in the detection of lipid-bound Hsp70.

In general, there are two main points of critique. Firstly, the authors can not convincingly show data for the detection of vesicular/liposomal Hsp70. Secondly, the manuscript lacks data showing the clinical utility and superiority over other liquid biopsy methods. Therefore, the title is misleading and can not include ‘exosomal’ and ‘response monitoring’ due to the lack of data to support it.

The overall assay and study design is interesting and might have important future implications for vesicle-based biomarker design. Nevertheless, important quality control parameters for the whole vesicular part are missing and are necessary to support the claims of the manuscript. If not available or possible to generate, the manuscript has to be toned down to circulating Hsp70 rather than lipid-bound. The platform EV track might give good indications on the required quality control data. It would be highly recommended to show:

1. Show the performance on isolated cancer vesicles with and without permeabilization.
2. Strengthen the clinical application of their approach eg by including more low-stage patients, correlate Hsp70 levels to therapy response, or progression-free/overall survival. Correlate the Hsp70 levels to tumor size?  

Abstract

Line 25 – Rephrase the term ‘microvesicles termed exosomes’ to the current nomenclature of extracellular vesicles. When using the term exosomes, it should be considered that this term historically described multivesicular body-derived vesicles that did not derive from the plasma membrane. With the suggested membrane localization of Hsp70 resulting in ‘vesicular release,’ the biogenesis pathway would be completely different

Line 25 – What indicates a conformational change of Hsp70 in the lipid environment? Can you show this? Is this published and could be used as a reference? In the current study, either denatured Hsp70 is used or native Hsp70 in combination with lipids which doesn’t allow any conclusions on this statement.

Line 42 – How do you show that it is a predictive biomarker? Do you not only show that it is a biomarker? How do your results allow for risk assessment? How could you use your data for monitoring therapeutic outcomes? Please specify or tone down.

Introduction

The introduction does not really put the presented data into perspective such as information about the state of the art of liquid biopsies in these entities is missing. The described clinical problems such as late diagnosis and poor treatment options are not addressed by the presented data and therefore irrelevant. It would be more informative to include information about the conformational changes of Hsp70 after membrane insertion? Whether the expression of HSP70 at the membrane of cells is specific to cancer or also a general marker for stress such as inflammation? Would the presented approach benefit from a combination with additional markers for cancer? What is known about the ratios of free versus membrane-bound Hsp70? What are the Hsp70 levels in early cancer stages? What are the sensitivities and detection rates for other liquid biopsies assays of NSCLC or HGG?

Line 59: How could the presented assay help to increase therapeutic success? How could the presented data increase the life expectancy of patients?Line 65: How does a drop in Hsp70 levels help to monitor therapeutic responses?

Line 95: How do normal cells Hsp70 levels change after exposure to environmental stress?

Line 99/100 : exosome is a term which should be updated to small Extracellular Vesicles (MISEV guidelines 2018).

Line 108 : Is the level of sEV-membrane HSP70 really more informative than free HP70 in the serum ? What about denaturing the proteins to make them work with the other antibodies?What about vesicle capturing with the described antibodies?

Results

Line 264: Reference missing. How much free Hsp70 is released by tumor cells?

Line 274 – 289: Most commercial antibodies recognize both HSP70 and HSC70 – if the antibody recognizes an epitope that is one AA different from HSC70 is it sufficient to make it specific? Can you show the discrimination?

Table 1: this is the AA sequence of the epitope recognized by the antibody, not the sequence of the antibody, it’s a bit misleading the title of the columns

Line 299 ff: Elaborate on this part of the data or integrate in another paragraph eg 3.3. Does this assay work on native, folded protein or on the denatured form? How do you explain that such similar binding affinities look so different on the western blot in figure 2a?

Figure 1. Resolution of the figures is poor, adjust scale bars of ligand concentration. Adjust a) and b) to top of the figure. What is the unit of the ligand concentration. Highlight assessed binding affinities with a line.

Line 311: Add western blot of the recombinant protein, otherwise it is misleading that you did it on viable tumor cells

Line 321: The percentage of positive cells does not match the data in the figure.

Figure 2a): Is there a way to include a loading control to strengthen the different antibody affinities? A western blot of cell lysates of the respective cell lines would be interesting to see the specificity of the antibody. Can you see a clear band or many different ones?

Figure 2b): The flow cytometry part as presented is weak. Is the experiment reproducible? Showing histograms is nice, but reproducing the experiment and making statistics would make the results stronger. How does it look on healthy, non-tumoral cells? Eg PBMCs would be highly recommended to prove the tumor specificity of the membrane-bound HSP70 What can explain the different ratios of Hsp70.1 and 2 between the two cell lines?

Figure 3: Units in []

Line 335 : liposomal HSP70 and exosomal HSP70 mean different things. The use of the term liposomal is misleading.

Line 347: Shouldn’t the calibration curve be measured on liposomal Hsp70 when this is the suggested biomarker application?

Line 359: LoB not introduced

Figure 4: The authors lack any proof of the suggested differentiation of liposomal vs free Hsp70. Potentially their two antibodies just recognize the most stable peptide of Hsp70, while Hsp70A binds to a peptide prone to protease cleavage. Does the ELISA work with Hsp70A or B after Triton treatment? Would a ratio of Hsp70.1 or .2 to Hsp70A or B give information about the actual vesicle bound fraction? What about immunoelectron microscopy of patients EVs to prove the surface labeling by Hsp70.1 and 2?

Figure 5: the reproducibility of the results between plasma and serum are good and enforce the assay’s strength suggested by the authors. However, once more, there is no proof that HSP70 recognized not exclusively the free HSP70 and not the exosomal HSP70 as the authors use plasma or serum directly.

Would people with inflammation or other diseases but not cancerous have higher levels of HSP70 as well that could lead to false-positive results?

Table 3/Figure 6: Why was the SEM used? Given that the approach is suggested as a biomarker for cancer detection, what about the large overlap between HGG patients and healthy volunteer Hsp70 concentration?

Figure 6

  • Please show patients individually, such as in box plots or dot plots What error is indicated in figure 1a?

C and D – same comment as in A.

D- Please indicate the treated entity in the figure legend. How can it be explained that the average before RT value is much higher than the stage III and stage IV Hsp70 concentration? What do we learn from the Hsp70 concentration during and after RT? Does it have any correlation with treatment response or prognosis? What is the time between last RT and sampling? Does RT itself cause an increase in Hsp70 levels in circulation?

Table 3 : What does the Threshold define?

Table 4: The data for stages III and IV look promising and convincing. Nevertheless, in the introduction, the authors claim that this could be used as a tool for early detection of cancer or minimal residual disease detection. With this table, they can not prove this point. In addition. What is the benefit of this assay over established cell-free DNA sequencing assays already published for NSCLC? Any improvement? How does the ELISA perform in terms of specificity and sensitivity to other NSCLC liquid biopsy assays?

Discussion

Line 496 : once more the difference between liposomal HSP70 and exosomal HSP70 is misleading.

Author Response

Reviewer 1

Comments and Suggestions for Authors

The authors of the manuscript ‘ Exosomal Hsp70 in liquid biopsies – a biomarker for prediction and response monitoring in cancer’ describe an ELISA-based method to detect cancer based on exosomal Hsp70. They describe a novel set of antibodies that are suggested to have superior performance over commercially available ones once in the detection of lipid-bound Hsp70.

In general, there are two main points of critique. Firstly, the authors can not convincingly show data for the detection of vesicular/liposomal Hsp70. Secondly, the manuscript lacks data showing the clinical utility and superiority over other liquid biopsy methods. Therefore, the title is misleading and can not include ‘exosomal’ and ‘response monitoring’ due to the lack of data to support it.

The overall assay and study design is interesting and might have important future implications for vesicle-based biomarker design. Nevertheless, important quality control parameters for the whole vesicular part are missing and are necessary to support the claims of the manuscript. If not available or possible to generate, the manuscript has to be toned down to circulating Hsp70 rather than lipid-bound. The platform EV track might give good indications on the required quality control data. It would be highly recommended to show:

  1. Show the performance on isolated cancer vesicles with and without permeabilization.

Answer

This point is well taken. Additional experiments have been performed and included as an additional Figure 2d. The data show the results of exosomal Hsp70 derived from EDTA blood of a tumor patient and a healthy human donor measured with the compHsp70 ELISA and a control ELIS detecting only free Hsp70. Exosomes were isolated by ultracentrifugation and were characterized by dynamic light scattering on a Zetasizer NanoS instrument. Exosomes were also isolated form supernatants of a tumor cell culture. The data were not included into the Ms, but again they demonstrate that exosomal Hsp70 can only be detected by the compHsp70 ELISA.

The exosomes were suspended in PBS and Stabilzyme Detect buffer which causes a permeabilization of the vesicles. Other detergents were also tested, however, these detergents negatively impacted on the ELISA results because they hindered the binding of Hsp70 to the ELISA plate. 

Figure: Exosomal Hsp70 derived from cell culture supernatant of a 60-70% confluent tumor cell line which expresses Hsp70 on the cell membrane (10 ml supernatant)

  1. Strengthen the clinical application of their approach eg by including more low-stage patients, correlate Hsp70 levels to therapy response, or progression-free/overall survival. Correlate the Hsp70 levels to tumor size?  

Answer

Since NSCLC as well as glioblastoma is diagnosed at a late tumor stage patients with low grade tumors in these entities are rare. We are collecting samples for future studies, however, presently we cannot provide more data for low grade tumor patients.

Previous data of our group indicate that Hsp70 levels in the circulation correlate with the gross tumor volume in NSCLC patients ([57] Gunther et al. Front 2015).

Abstract

Line 25 – Rephrase the term ‘microvesicles termed exosomes’ to the current nomenclature of extracellular vesicles. When using the term exosomes, it should be considered that this term historically described multivesicular body-derived vesicles that did not derive from the plasma membrane. With the suggested membrane localization of Hsp70 resulting in ‘vesicular release,’ the biogenesis pathway would be completely different

Answer

This aspect is well taken the term “microvesicles” was substituted by “extracellular vesicles”. Only when artificial unilamellar lipid vesicles were meant the term artificial lipid vesicle was used.

Moreover, we have published previously that exosomes (small extracellular vesicles) derived from membrane Hsp70-positive tumor cells show biophysical characteristics of exosomes (small extracellular vesicles) ([27] Gastpar et al Can Res 2005).

Line 25 – What indicates a conformational change of Hsp70 in the lipid environment? Can you show this? Is this published and could be used as a reference? In the current study, either denatured Hsp70 is used or native Hsp70 in combination with lipids which doesn’t allow any conclusions on this statement.

Answer

An interaction of Hsp70 with lipids (i.e. PS, Gb3, sphingolipids) in the membrane causes conformational changes as outlined in different publications which are cited in the Ms (Smulders et al IJMS 2020; Balogi et al Progress in lipid Res 2019). The antibodies cmHsp70.1 (Stangl et al PNAS 2011) and cmHsp70.2 (Figure 2b) are able to detect exosomal/vesicular and membrane-bound Hsp70 as well as free Hsp70, as demonstrated by flow cytometric analysis of viable tumor cells and by Western blotting, whereas other control Hsp70 antibodies (ctrl Hsp70A/B) are only detecting free and denatured Hsp70 in aqueous solutions, but not membrane-bound Hsp70 on the cell surface of viable tumor cells and exosomes (Figure 2b).

Line 42 – How do you show that it is a predictive biomarker? Do you not only show that it is a biomarker? How do your results allow for risk assessment? How could you use your data for monitoring therapeutic outcomes? Please specify or tone down.

Answer

We have shown in the Ms that Hsp70 values in grade IV NSLCL (squamous as well as adeno carcinoma) have higher Hsp70 values in the circulation than grade III NSCLC. We also have shown in this study that the Hsp70 values dropped in patients who responded to therapy (radiation therapy). A correlation of extracellular Hsp70 and the gross tumor volume (GTV) has been demonstrated for NSCLC previously (Gunther et al. Front Immunol 2015). According to the comment of the reviewer we deleted the term “predictive”.  

Introduction

The introduction does not really put the presented data into perspective such as information about the state of the art of liquid biopsies in these entities is missing. The described clinical problems such as late diagnosis and poor treatment options are not addressed by the presented data and therefore irrelevant. It would be more informative to include information about the conformational changes of Hsp70 after membrane insertion? Whether the expression of HSP70 at the membrane of cells is specific to cancer or also a general marker for stress such as inflammation? Would the presented approach benefit from a combination with additional markers for cancer? What is known about the ratios of free versus membrane-bound Hsp70? What are the Hsp70 levels in early cancer stages? What are the sensitivities and detection rates for other liquid biopsies assays of NSCLC or HGG?

Answer

These comments are well taken and the Introduction was modified accordingly.

The ratio of free and membrane-bound Hsp70 is mentioned in the Discussion part.

Additional biomarkers in liquid biopsies which might further improve the results derived with the compHsp70 ELISA are also included in the Discussion.

Apart from serum and plasma the liquor of patients with glioblastoma has been tested previously. However, the amount of Hsp70 was below the detection level of the compHsp70 ELISA. Therefore, no data can be provided on liquor. Other liquid biopsies have not yet been tested.

Line 59: How could the presented assay help to increase therapeutic success? How could the presented data increase the life expectancy of patients? Line 65: How does a drop in Hsp70 levels help to monitor therapeutic responses?

Answer

The present approach could help to increase therapeutic success since blood samples can be taken more frequently by a minimal invasive method during therapy and thereby allows a continuous monitoring of tumor responses (as determined by a drop/increase in circulating Hsp70 values). These data could result (after confirmation with standard imaging methods such as MRI, PET-CT, etc.) in a faster therapy adaptation and thereby help to increase therapeutic success (see Introduction).

Line 95: How do normal cells Hsp70 levels change after exposure to environmental stress?

Answer

Upon stress (i.e. a sublethal heat shock, cytostatic drugs) the Hsp70 levels increase in the cytosol of normal cells, but Hsp70 does not get translocated to the cell surface ([20]Gehrmann et al Biol Chem 2002). Since exosomes reflect the protein (Hsp70) composition of the plasma membrane no increase in Hsp70 in the extracellular milieu is found by stressed exosomes derived from normal cells.

Line 99/100 : exosome is a term which should be updated to small Extracellular Vesicles (MISEV guidelines 2018).

Answer

The term exosome was updated to “small extracellular vesicles” according to the MISEV guidelines 2018.

Line 108 : Is the level of sEV-membrane HSP70 really more informative than free HP70 in the serum ? What about denaturing the proteins to make them work with the other antibodies? What about vesicle capturing with the described antibodies?

Answer

As mentioned in the discussion the values of sEV Hsp70 are approximately 100-fold higher than that of free Hsp70. Moreover, the values of free Hsp70 did not reveal any significant differences before and after radiotherapy.

Although the point is well taken we could show that a denaturation of exosomes with other detergents did not result in an increase of free Hsp70 since the denaturation agents negatively affected the binding capacity of free Hsp70 to the Hsp70 ELISA antibodies.

However, we already have demonstrated that circulating tumor cells CTCs (even after EMT) expressing Hsp70 on their cell surface can be captured from the blood of tumor patients by using the cmHsp70.1/2 antibodies (Breuninger et al Front Oncol 2018). Therefore, we assume that it would be also possible to use the antibodies for capturing extracellular vesicles presenting Hsp70 on their surface. 

Results

Line 264: Reference missing. How much free Hsp70 is released by tumor cells?

Answer

A reference was included. As mentioned in the discussion the amount of free Hsp70 was approximately 100-times lower than that of liposomal Hsp70. The exact values of free Hsp70 of tumor patients have been included into the results part. The differences in the free Hsp70 values before and after therapy did not differ significantly.

Line 274 – 289: Most commercial antibodies recognize both HSP70 and HSC70 – if the antibody recognizes an epitope that is one AA different from HSC70 is it sufficient to make it specific? Can you show the discrimination?

Answer

For this discrimination dot blot analysis have been performed with recombinant Hsp70, Hsc70 and Hsp60. The data clearly indicate that the antibodies cmHsp70.1/2 and ctrl Hsp70A are only recognizing Hsp70 and do not cross-react with Hsc70. Recombinant Hsp60 as a control was not recognized by any of the 4 Hsp70 antibodies. In contrast, the ctrl Hsp70B antibody recognizes both Hsp70 and Hsc70. A representative dot blot analysis using ctrlHsp70B and cmHsp70.1 has been included as a new Figure 2c.

Table 1: this is the AA sequence of the epitope recognized by the antibody, not the sequence of the antibody, it’s a bit misleading the title of the columns.

Answer

This is correct it is the sequence of Hsp70 (HSPA1A) in the different species which is recognized as an epitope recognized by the antibodies. This has been clarified in Table 1 and in the results part.

Line 299 ff: Elaborate on this part of the data or integrate in another paragraph eg 3.3. Does this assay work on native, folded protein or on the denatured form? How do you explain that such similar binding affinities look so different on the western blot in figure 2a?

Answer

The binding affinities (KD) of cmHsp70.1 and cmHsp70.2 mAbs to recombinant Hsp70 protein, as determined by MST measurements, are 0.42 and 0.44 nM, respectively. All 4 antibodies detect denatured Hsp70, as determined by Western blot analysis of tumor cell lysates after SDS page as well as recombinant non-denatured Hsp70 protein, as determined by dot blot analysis of recombinant HSP proteins.

The differences in the intensity of the bands for cmHsp70.1 and cmHsp70.2 are most likely due to the fact that the secondary antibody for cmHsp70.1 is a mouse antibody whereas that for cmHsp70.2 is a rat antibody.

Figure 1. Resolution of the figures is poor, adjust scale bars of ligand concentration. Adjust a) and b) to top of the figure. What is the unit of the ligand concentration. Highlight assessed binding affinities with a line.

Answer

The labeling has been adjusted. The resolution has been improved and the binding affinity was highlighted with an arrow as recommended.

Line 311: Add western blot of the recombinant protein, otherwise it is misleading that you did it on viable tumor cells

Answer

The wording “of recombinant Hsp70” has been included. Moreover, Western blot analysis of tumor cell lysates have been included as well. 

Line 321: The percentage of positive cells does not match the data in the figure.

Answer

This was a typing error in the text. Now mean data of three independent FACS experiments have been shown in the results part, as recommended by the reviewer.

Figure 2a): Is there a way to include a loading control to strengthen the different antibody affinities? A western blot of cell lysates of the respective cell lines would be interesting to see the specificity of the antibody. Can you see a clear band or many different ones?

Answer

Western blot data of tumor cell lysates together with loading controls (actin) have been included into Figure 2a as suggested. Only 1 clear Hsp70 band is visible with the antibodies cmHsp70.1, cmHsp70.2 and ctrl Hsp70A and therefore it was concluded that these antibodies are specific for Hsp70 and do not cross-react with Hsc70. The antibody ctrl Hsp70B shows a faint additional band at 73 kDa which represents Hsc70. This finding was confirmed by dot blot analysis using recombinant Hsc70 as an antigen (Figure 2c).

In the Western blot a loading control was included, for the recombinant Hsp70 protein this is not possible.

Figure 2b): The flow cytometry part as presented is weak. Is the experiment reproducible? Showing histograms is nice, but reproducing the experiment and making statistics would make the results stronger. How does it look on healthy, non-tumoral cells? Eg PBMCs would be highly recommended to prove the tumor specificity of the membrane-bound HSP70. What can explain the different ratios of Hsp70.1 and 2 between the two cell lines?

Answer

The mean values of at least three independent flow cytometric experiments have been included into the results part to strengthen the data shown in Figure 2b. Moreover, a FACS analysis of PBL of healthy human donors has been included as an additional Figure. The mean values of n=5 independent experiments have been included.

Figure 3: Units in []

Answer

The OD has no dimension, this was corrected in the Figure.

Line 335 : liposomal HSP70 and exosomal HSP70 mean different things. The use of the term liposomal is misleading.

Answer

Liposomal was now used only when artificial liposomes were mentioned.

Line 347: Shouldn’t the calibration curve be measured on liposomal Hsp70 when this is the suggested biomarker application?

Answer

Calibration curves with liposomal Hsp70 were performed, however, due to the formation of micelles of the liposomes the standard curves did vary. Therefore, only recombinant Hsp70 was used as a standard.

Line 359: LoB not introduced

Answer

LoB means Limit of Blank (has been included)

Figure 4: The authors lack any proof of the suggested differentiation of liposomal vs free Hsp70. Potentially their two antibodies just recognize the most stable peptide of Hsp70, while Hsp70A binds to a peptide prone to protease cleavage. Does the ELISA work with Hsp70A or B after Triton treatment? Would a ratio of Hsp70.1 or .2 to Hsp70A or B give information about the actual vesicle bound fraction? What about immunoelectron microscopy of patients EVs to prove the surface labeling by Hsp70.1 and 2?

Answer

EM data of tumor derived exosomes using the antibody cmHsp70.1 have been provided in a previous publication ([27] Gastpar et al Can Res 2005) which is cited in the Ms. The biophysical characteristics of exosomes (such as AChe activity, density, protein composition) have been demonstrated in this publication. In the present work the exosomes isolated from the plasma were characterized by dynamic light scattering on a Zetasizer instrument as shown in the upper part of a new Figure 2d. We could show that exosomal Hsp70 was only detectable with the comHsp70 ELISA, but not with a control Hsp70 ELISA detecting only free Hsp70. Moreover, exosomal Hsp70 derived from the cell culture supernatant of tumor cells revealed similar results (see below). This data have not been included into the Ms. 

Exosomal Hsp70 derived from cell culture supernatant of a 60-70% confluent tumor cell line which expresses Hsp70 on the cell membrane (10 ml supernatant)

Following treatment of the exosomes with detergents the binding of Hsp70 in the ELISA was negatively affected.

Figure 5: the reproducibility of the results between plasma and serum are good and enforce the assay’s strength suggested by the authors. However, once more, there is no proof that HSP70 recognized is not exclusively the free HSP70 and not the exosomal HSP70 as the authors use plasma or serum directly.

Answer

In addition to artificial liposomes with recombinant Hsp70 protein, we have isolated exosomes from EDTA blood of a tumor patient and a healthy human individual. We could show that the compHsp70 ELISA detects exosomal Hsp70 isolated from the plasma of a tumor patient but not from a healthy human donor. Moreover, a control Hsp70 ELISA detecting only free Hsp70 was unable to detect exosomal Hsp70 isolated from the plasma.

Would people with inflammation or other diseases but not cancerous have higher levels of HSP70 as well that could lead to false-positive results?

Answer

A comparison of extracellular Hsp70 levels in serum of patients with chronic hepatitis, liver cirrhosis and hepatocellular carcinoma revealed significantly higher Hsp70 levels in carcinoma patients compared to patients with inflammation [30]. This finding was mentioned in the introduction.

Table 3/Figure 6: Why was the SEM used? Given that the approach is suggested as a biomarker for cancer detection, what about the large overlap between HGG patients and healthy volunteer Hsp70 concentration?

Answer

The SD values have been included into the box blot Figures.

Figure 6

  • Please show patients individually, such as in box plots or dot plots What error is indicated in figure 1a?

C and D – same comment as in A.

Answer

As recommended box plots have been included as suggested. SD values are shown.

D- Please indicate the treated entity in the figure legend. How can it be explained that the average before RT value is much higher than the stage III and stage IV Hsp70 concentration? What do we learn from the Hsp70 concentration during and after RT? Does it have any correlation with treatment response or prognosis? What is the time between last RT and sampling? Does RT itself cause an increase in Hsp70 levels in circulation?

Answer

The patients included into the study showed a treatment response after RT. This information is mentioned in the results part.

RT alone does increase the values of free Hsp70 due to inflammation which lasts approximately 3 months after RT. The sampling of the blood was performed directly after the last irradiation. In the discussion it was mentioned that RT induced inflammation at the end of therapy might be responsible for the slight increase in Hsp70.

Table 3 : What does the Threshold define?

Answer

The threshold defines the difference between healthy and tumor patients in a specific tumor entity.

Table 4: The data for stages III and IV look promising and convincing. Nevertheless, in the introduction, the authors claim that this could be used as a tool for early detection of cancer or minimal residual disease detection. With this table, they can not prove this point. In addition. What is the benefit of this assay over established cell-free DNA sequencing assays already published for NSCLC? Any improvement? How does the ELISA perform in terms of specificity and sensitivity to other NSCLC liquid biopsy assays?

Answer

The authors agree that early tumor detection has not been proven by the data shown in the Ms. The statement has been deleted in the Introduction as recommended.

Previous studies have indicated that the gross tumor volume correlates with Hsp70 values in the circulation (Gunther et al Front 2015). This study has been included into the Ms. Moreover, we could demonstrate that a combination of Hsp70 values with the hypoxia-related marker osteopontin can better predict clinical responses in NSCLC patients (Ostheimer et al Front 2017).

Discussion

Line 496 : once more the difference between liposomal HSP70 and exosomal HSP70 is misleading.

Answer

The use of liposomal and exosomal has been corrected and unified. Additional data have been included showing results with exosomes isolated from EDTA blood of a patient and a healthy individual.  

The authors want to thank both reviewers for constructive and helpful comments and suggestions.

Reviewer 2 Report

In general this is a nice manuscript. It is well written, the design of experiments is logic and in the most relevant experiments  the number of patient sample is not too small to end up with reliable results. That said, I do have some remarks that should to be addressed prior publication:

Major remarks:

  • R400 - The author states that compHsp70 ELISA results are not significantly different in serum and plasma based on the comparison of 13 healthy volunteers (low compHsp70 levels).  Testing a handful of negative/low positive samples is not sufficient to come to this conclusion. The comparison should be expanded to a larger amount of samples and to samples with a broad range of compHsp70 levels (high positives, intermediate positive and weak positives/negatives).

  • The validation (incl LOB and LOD calculation!!) should be performed for each matrix (plasma ó serum) separately. Now, the validation of serum samples is present but the validation of plasma samples is lacking.

  • R503 – “The compHsp70 ELISA allows the quantification of tumor-derived Hsp70 in serum and plasma with high precision and linearity in clinically relevant concentration range” – 2 remarks:
    • Insufficiently proven in this manuscript that Hsp70 in plasma of cancer patients is valuable since only (?) serum samples of cancer patients were tested.
    • High precision? Sensitivity of 68% in NSCLC and specificity of 33% in glioma patients is in contrast with this claim.

Minor remarks:

  • R245 - Interassay precision: were following variables introduced during these experiments: different operators, different days, different reagent lots (of key elements such as the antibodies)?
  • R444 “ Compared to healthy individuals (35 ± 3.99 ng/ml; n = 108), serum Hsp70 levels were significantly higher in patients with stage III and IV squamous cell carcinoma of the lung comparing patient results with healthy volunteers” Is this comparison performed using samples of same matrix of is this comparison based on a mix of plasma and serum samples ? If the latter, than the comparison should be restricted to samples of same matrici (as currently insufficient prove of similar results with both matrici is demonstrated).
  • LOB = mean blank + 1.645(SDblank). What is the rational of using 1.645 instead of 2 or even 3x SD? This formula with 1.645 leads to lower specificity (or more false positive results). Idem ditto question for LoD formula.
  • CV ranges between 0.02% and 12.5%. Is this depending on the compHsp70 levels or on the input concentration (high CV at low conc or low compHsp70 levels and low CV at high concentration/compHsp70 levels)? Or is there another explanation for this broad range?
  • The % CV can be quit high (up to 12.5%). Did you test the patient samples in duplicate or triplicate to compensate for this variation in quantification? (A method to reduce the variation in quantification should largely benefit the results.)
  • Why testing the influence of age and food intake and not the influence of other parameters such as physical activity and inflammation?
  • Table 3 – the sensitivity (68% and 91%) and specificity (94% and 33%) in lung and glioma should be discussed in more detail as not all results (e.g. specificity of 33%) are promising.

Author Response

Reviewer 2

Comments and Suggestions for Authors

In general this is a nice manuscript. It is well written, the design of experiments is logic and in the most relevant experiments  the number of patient sample is not too small to end up with reliable results. That said, I do have some remarks that should to be addressed prior publication:

Major remarks:

  • R400 - The author states that compHsp70 ELISA results are not significantly different in serum and plasma based on the comparison of 13 healthy volunteers (low compHsp70 levels).  Testing a handful of negative/low positive samples is not sufficient to come to this conclusion. The comparison should be expanded to a larger amount of samples and to samples with a broad range of compHsp70 levels (high positives, intermediate positive and weak positives/negatives).

Answer

We included a total of 108 samples of healthy donors. Among the 108 samples most Hsp70 values were in a similar low range. For the comparison of serum and plasma samples healthy donors were chosen whose Hsp70 values were high, low and intermediate as recommended by the reviewer. Out of the 108 samples there were not too many samples with these characteristics, therefore the number of serum and plasma samples was pretty low.  

  • The validation (incl LOB and LOD calculation!!) should be performed for each matrix (plasma ó serum) separately. Now, the validation of serum samples is present but the validation of plasma samples is lacking.

Answer

The validation has been performed for both plasma and serum with nearly identical values.

  • R503 – “The compHsp70 ELISA allows the quantification of tumor-derived Hsp70 in serum and plasma with high precision and linearity in clinically relevant concentration range” – 2 remarks:
    • Insufficiently proven in this manuscript that Hsp70 in plasma of cancer patients is valuable since only (?) serum samples of cancer patients were tested.

Answer

The authors agree that more plasma samples of patients need to be tested. However, due to time restrictions we were not able to obtain more plasma samples. Plasma samples from a patient was used to isolate exosomes. We could demonstrate that Hsp70 from plasma derived exosomes could be quantified with the compHsp70 ELISA but not with a control ELISA that detects only free Hsp70.

High precision? Sensitivity of 68% in NSCLC and specificity of 33% in glioma patients is in contrast with this claim.

Answer

A comparison of specificities of presently used biomarkers for NSCLC reveal that the specificity of Hsp70 as a biomarker for NSCLC and HGG are in the normal range. “e.g., cytokeratin 19 fragment (CYFRA 21-1, specificity 76%), carcinoma embryonic antigen (CEA, specificity 52%), carbohydrate antigen 125 (CA125, specificity 52%), carbohydrate antigen 153 (CA153), carbohydrate antigen 199 (CA199), neuron-specific enolase (NSE, specificity 22%)”

Minor remarks:

  • R245 - Interassay precision: were following variables introduced during these experiments: different operators, different days, different reagent lots (of key elements such as the antibodies)?

Answer

The assay was performed by at least 5 skilled operators, on different days and with at least three different reagent lots. This information has been included. 

  • R444 “ Compared to healthy individuals (35 ± 3.99 ng/ml; n = 108), serum Hsp70 levels were significantly higher in patients with stage III and IV squamous cell carcinoma of the lung comparing patient results with healthy volunteers” Is this comparison performed using samples of same matrix of is this comparison based on a mix of plasma and serum samples ? If the latter, than the comparison should be restricted to samples of same matrici (as currently insufficient prove of similar results with both matrici is demonstrated).

Answer

The assay was compared with samples of same matrici (only serum samples were compared from healthy individuals and tumor patients. This information has been included.

LOB = mean blank + 1.645(SDblank). What is the rational of using 1.645 instead of 2 or even 3x SD? This formula with 1.645 leads to lower specificity (or more false positive results). Idem ditto question for LoD formula.

Answer

LoB and LoD were established according to the Clinical Laboratory Standards Institute (CLSI) guideline EP17-A as summarized by Armbruster and Pry, 2008. Armbruster and Pry used this equation because „some LoD sample values are expected to be less than the estimated, but when using 1.645 SD, no more than 5% of the values were less than the LoB. If the observed LoD sample values meet this criterion, the LoD is considered established or verified.”

  • CV ranges between 0.02% and 12.5%. Is this depending on the compHsp70 levels or on the input concentration (high CV at low conc or low compHsp70 levels and low CV at high concentration/compHsp70 levels)? Or is there another explanation for this broad range?

Answer

The broad range is due to the fact that the compHsp70 ELISA measures very high Hsp70 levels in the serum and plasma of tumor patients which result in high CV values.

  • The % CV can be quit high (up to 12.5%). Did you test the patient samples in duplicate or triplicate to compensate for this variation in quantification? (A method to reduce the variation in quantification should largely benefit the results.)

Answer

The samples were tested in duplicates

  • Why testing the influence of age and food intake and not the influence of other parameters such as physical activity and inflammation?

Answer

Food intake was tested since it is important for future clinical applications whether the patient is allowed or not allowed to eat before blood sampling. Inflammation has been tested in previous studies. A comparison of extracellular Hsp70 levels in serum of patients with chronic hepatitis, liver cirrhosis and hepatocellular carcinoma revealed significantly higher Hsp70 levels in carcinoma patients compared to patients with inflammation [30]. This was mentioned in the introduction.

  • Table 3 – the sensitivity (68% and 91%) and specificity (94% and 33%) in lung and glioma should be discussed in more detail as not all results (e.g. specificity of 33%) are promising.

Answer

The reason for the lower specificity in glioblastoma might be due to the fact that the Hsp70 values in HGG are generally lower than in NSCLC. The blood brain barrier might be responsible for the lower Hsp70 values in the circulation.

A comparison of specificities of presently used biomarkers for NSCLC reveal that the specificity of Hsp70 as a biomarker for NSCLC and HGG are in a normal range. “e.g., cytokeratin 19 fragment (CYFRA 21-1, specificity 76%), carcinoma embryonic antigen (CEA, specificity 52%), carbohydrate antigen 125 (CA125, specificity 52%), carbohydrate antigen 153 (CA153), carbohydrate antigen 199 (CA199), neuron-specific enolase (NSE, specificity 22%)”

The authors want to thank both reviewers for constructive and helpful comments and suggestions.

Round 2

Reviewer 1 Report

The quality of the revised version of the manuscript does not allow for a scientific review process due to major inconsistencies and lacking information. I refrain from working on this version.  

Author Response

Dear Editorial board

we have adressed adequately all comments of both reviewers in the revised version of our Ms and we do not detect any inconsistencies or lack of information in the resubmitted version of our Ms. All new methods of the new data which were included into the Ms have been described in the Methods part.

The Ms has been proof-read by a native English speaker. The resubmitted version was again checked for spelling errors. Please find enclosed a re-revised version of our Ms.

All the best Gabriele Multhoff

Round 3

Reviewer 1 Report

I have expressed my concerns regarding this article in the attached document. They havent addressed the main point of proving the identity of extracellular vesicles. Please see the attachment for details.

Author Response

Point-by-point letter

Comment: "Vesicular Hsp70 serves as a tumor-specific biomarker" ...recommended to be replaced by “circulating Hsp70...”

Answer: Additional data (new Figures 6e,f) have been included that provide biophysical and biochemical evidence that the compHsp70 ELISA recognizes exosomal Hsp70 in tumor patients and supernatants of tumor cells but not a control Hsp70 ELISA.

Introduction: no significant difference between inflammation and tumor.

Answer: There is a misunderstanding. In the Ms Gehrmann et al Front Immunology (doi 10.3389/fimmu.2014.00307) significant differences in soluble Hsp70 values were shown for HCC (hepatocellular carcinoma) and healthy controls and patients with liver cirrhosis and patients with chronic hepatitis. 

Materials and Methods:

Information about tumor lysates are missing.

Answer: Information about tumor cell lysates have been provided as recommended.

Manufacturer Information is missing for cellulose membrane, horseradish peroxidase, diaminobenzidine.

Answer: The missing Manufacturer informations have been included.

FCS vesicle-depleted or how was the contamination with FCS-derived vesicles achieved.

Answer: In the plasma of patients no FCS is present. In case of exosomes derived from supernatant of tumor cell cultures fresh medium with FCS (10%) was used as a control and these values were subtracted. This information has been included as well. 

Uniprot comparison is missing:

Answer: Uniprot analysis has been described in the Materials and Methods part.

“Inter-species comparison of the 8-mer (aa 454-461) and 10-mer (aa 614-623) sequences of Hsc70 (HSPA8) and Hsp70 (HSPA1A) in humans were compared to the respective sequence of Hsp70 in mouse, rat, dog, bovine, horse, pig, and zebrafish by UniProtKB 2021_03 analysis of UniProt. In cats only the sequence of HSPA2 is available.”

Figure2: The Figure layout should be rearranged.

Answer: The complete Figure 2 was rearranged.

Figure 2a: Figure legend is not adopted.

Answer: The 40 kDa marks the position of the loading control actin. The Figure 2 as a total and the Figure legend were adopted, accordingly.

Comment: Figure 2c: Is there a reason why the dot plot is only shown for cmHsp70.1 and not cmHsp70.2.

Answer: A dot plot for cmHsp70.2 has been included. Additional protein controls such as BSA and Hsp27 have been included.

Comment: Figure 2d: A specification of the zetaview profile would be beneficial for tumor vs healthy. Which sample was measured.

Answer: These aspects have been addressed. In the revised version exosomes derived from a tumor patient and a healthy human donor were included and zetaview profiles have been shown in a new Figure 6e.

Comment Line 422: Can you use the same calibration curve for healthy and tumor patients since all tumor patients are above 100 ng/ml.

Answer: For tumor patients the same standard curves were used than for healthy human donors with the highest value of 100 ng/ml, but the serum samples of tumor patients were diluted 1:20 instead of 1:5 in dilution buffer when the Hsp70 values of tumor patients were too high.

Paragraph 3.5: Could you show the quantification and recovery both for lipid bound and free Hsp70. To show the specific antibody performance on the vesicular fraction?

Answer: A quantification and recovery of free Hsp70 and lipid-bound Hsp70 was performed in buffer and plasma. The data of the free Hsp70 in buffer and plasma are summarized in Table 2. The data of lipid-bound Hsp70 are shown in Figure 4.

Table 4: What statistical test was used.

Answer: Data of all groups were compared using a one-way analysis of variance (ANOVA) with Tukey test for multiple comparisons in RStudio. The pairwise comparison of groups was performed with a 2-sided t-test. The respective test for each comparison has been included in the manuscript.

Comment Line 537:... responding and non-responding individuals need to be included

Answer: This is correct but since we do not have the samples available this aspect could not be addressed in this Ms. 

The wording predictive has been deleted from the title as recommended.

Comment Figure 6c/d: ...statistics should be calculated between SCC III and IV and Adeno ca III and IV. …which t-test was used.

Answer: Hsp70 serum levels in Adeno ca III and IV differed statistical significant (p<0.05; ANOVA Tukey test). These data have been included into Figure 6.

Comment Line 591: liquid biopsy test for NSCLC based on cfDNA and sensitivities need to be included. They have been proven to be informative under EGFR inhibitor treatment. In which sense is the ELISA better.

Answer: Detection of the circulating tumor cell (ctDNA) and cell free DNA (cfDNA) represents one of the approaches for the assessment of the tumor progression and response to the provided therapy. However, there are certain limitations when cfDNA is applied as a tumor biomarker. A major limitation is due to false negative result which can occur because of many different reasons, including the low signal-to-noise ratio, and the short half-life (<1.5 hours) of ctDNA (Bettegowda et al., 2014). When soluble Hsp70 is used as a biomarker tumor progression can be assessed because Hsp70 is predominantly released by viable tumor cells and therefore, Hsp70 might serve as biomarker for the viable tumor mass (Gunther et al 2017). Furthermore, the half-life of circulating of Hsp70 (particularly under stress conditions) is much longer (in the range of 7 hours) than that of ctDNA (< 1.5 h) (Li et al., 1995). In this regard soluble Hsp70 provides a relative stable tumor biomarker in the blood which enables the assessment of the viable tumor mass and therefore might be able to predict tumor responses. In mouse models soluble Hsp70 levels correlated with the viable tumor mass and predicted therapy responses to radiotherapy (unpublished observation)

Comment: What is the source of the 100-fold more Hsp70 than free Hsp70.

Answer: The values derived for free Hsp70 has been shown in the results part on page 15 (it is indicated in yellow).

Comment: highlight the novelty of this study

Answer: This novel ELISA is based on two monoclonal antibodies whereas the old ELISA was based on one monoclonal antibody and a rabbit antiserum. Since the Hsp70 antiserum varied significantly between different immunizations of the rabbits the previous ELISA did not provide stable data. Apart from a much better reproducibility of the results the sensitivity of the ELISA is higher.  

Data of Hsp70 values derived from exosomes (tumor patient, healthy human individual, cell culture supernatant) have been included.

Comment line 633: rephrase the wording clear discrimination.

Answer: This comment was rephrased to "discrimination"

Comment line 642: Significance after RT is not shown.

Answer: After receiving a radiation dose of approximately 20 Gy (range 18 – 22.5 Gy) Hsp70 levels dropped significantly. After completion of radiotherapy (60 – 70 Gy) the Hsp70 levels remained at a low level.

Comment to line 644: no data for the commercial ELISA on this sample:

Answer: The data are included in the Results part on page 15.

Comment line 677: Are there any good therapeutic adaptations possible in HGG?

Answer: Inhibitors of the isocitrate dehydrogenase (IDH) represent promising approaches in the treatment of HGG. Preclinical studies demonstrated that inhibitors of mutant (mIDH) -1 and -2 prevent the accumulation of the oncometabolite d-2-hydroxyglutarate (2-HG) and therefore could be beneficial for the treatment of glioblastoma (Clark et al., 2016). In a recent study Vorasidenib (AG-881) has been shown to inhibit the production of 2-HG in glioma tissue by >97% in an orthotopic glioma mouse model (Konteatis et al., 2020). Furthermore, a case report from a phase I study showed improved seizure control and radiographic stable disease in glioblastoma patient for more than 4 years by a treatment with ivosidenib (Tejera et al., 2020). Another approach based on IDH mutation in glioblastomas was suggested by Platten et al. by employing an IDH1(R132H)-specific peptide vaccine (IDH1-vac) that induces specific therapeutic T helper cell responses that are effective against IDH1(R132H)+ tumors in preclinical and first-in-humans phase I trial (Platten et al., 2021).

New References included into the Ms

  1. Clark O, Yen K, Mellinghoff IK. Molecular Pathways: Isocitrate Dehydrogenase Mutations in Cancer. Clin Cancer Res. 2016 Apr 15;22(8):1837-42. doi: 10.1158/1078-0432.CCR-13-1333. Epub 2016 Jan 27. PMID: 26819452; PMCID: PMC4834266.
  2. Konteatis Z, Artin E, Nicolay B, Straley K, Padyana AK, Jin L, Chen Y, Narayaraswamy R, Tong S, Wang F, Zhou D, Cui D, Cai Z, Luo Z, Fang C, Tang H, Lv X, Nagaraja R, Yang H, Su SM, Sui Z, Dang L, Yen K, Popovici-Muller J, Codega P, Campos C, Mellinghoff IK, Biller SA. Vorasidenib (AG-881): A First-in-Class, Brain-Penetrant Dual Inhibitor of Mutant IDH1 and 2 for Treatment of Glioma. ACS Med Chem Lett. 2020 Jan 22;11(2):101-107. doi: 10.1021/acsmedchemlett.9b00509. PMID: 32071674; PMCID: PMC7025383.

70.Tejera D, Kushnirsky M, Gultekin SH, Lu M, Steelman L, de la Fuente MI. Ivosidenib, an IDH1 inhibitor, in a patient with recurrent, IDH1-mutant glioblastoma: a case report from a Phase I study. CNS Oncol. 2020 Sep 1;9(3):CNS62. doi: 10.2217/cns-2020-0014. Epub 2020 Jul 27. PMID: 32716208; PMCID: PMC7546125.

71.Platten M, Bunse L, Wick A, Bunse T, Le Cornet L, Harting I, Sahm F, Sanghvi K, Tan CL, Poschke I, Green E, Justesen S, Behrens GA, Breckwoldt MO, Freitag A, Rother LM, Schmitt A, Schnell O, Hense J, Misch M, Krex D, Stevanovic S, Tabatabai G, Steinbach JP, Bendszus M, von Deimling A, Schmitt M, Wick W. A vaccine targeting mutant IDH1 in newly diagnosed glioma. Nature. 2021 Apr;592(7854):463-468. doi: 10.1038/s41586-021-03363-z. Epub 2021 Mar 24. PMID: 33762734; PMCID: PMC8046668.

Comment: What about proteasome treatment to prove the identity on a vesicular surface comparted to luminal.

Answer: We did test different proteases on the samples but when we applied the samples to the ELISA plate the binding of the Hsp70 to the sandwich ELISA was inhibited, meaning the values dropped. 

Comment: What does the sentence refer to:

Answer: The sentence referred to ELISA results of exosomes isolated from the supernatant of tumor cell cultures. These data have been included into the Ms.

Comment: It is hard to believe that there is no Hsp70 detected in the CSF.

Answer: Hsp70 was detectable in the CSF, but at low levels. This has been included.

Comment: where are the values for free Hsp70.

Answer: The values are described in the Results part on page 15.

Comment: Where is the actin loading control.

Answer: The actin loading control (at 40 kDa) has been included into the main text and also into the Figure legend, as recommended.

The authors want to thank the reviewers for constructive comments.